# Insensitivity of alkenone carbon isotopes to atmospheric CO₂ at low to moderate CO₂ levels

Marcus P.S. Badger*[1,2,] Thomas B. Chalk[3,4], Gavin L. Foster[3], Paul R. Bown[5], Samantha J. Gibbs[3], Philip F. Sexton[1], Daniela N. Schmidt[6,7], Heiko Pälike[8], Andreas Mackensen[9] and Richard D. Pancost[2,7]

[1]School of Environment, Earth & Ecosystem Sciences, The Open University, Milton Keynes, MK7 6AA, UK
[2]Organic Geochemistry Unit, School of Chemistry, School of Earth Sciences, University of Bristol, Bristol, BS8 1TS, UK
[3]School of Ocean and Earth Science, National Oceanography Centre Southampton, University of Southampton, Southampton SO14 3ZH, UK
[4]Department of Physical Oceanography, Woods Hole Oceanographic Institution, Woods Hole, MA, 02543, USA
[5]Department of Earth Sciences, University College London, London, WC1E 6BT, UK
[6]School of Earth Sciences, University of Bristol, Wills Memorial Building, Queens Road, Bristol, BS8 1RJ, UK
[7]The Cabot Institute, University of Bristol, Bristol BS8 1UJ, UK
[8]MARUM – Center for Marine Environmental Sciences, University of Bremen, Bremen, Germany
[9]Alfred Wegener Institute for Polar and Marine Research, Am Alten Hafen 26, 27568 Bremerhaven, Germany

Correspondence to: Marcus P.S. Badger (marcus.badger@open.ac.uk)

**Abstract** Atmospheric $p$CO₂ is a critical component of the global carbon system and is considered to be the major control of Earth's past, present and future climate. Accurate and precise reconstructions of its concentration through geological time are, therefore, crucial to our understanding of the Earth system. Ice core records document $p$CO₂ for the past 800 kyrs, but at no point during this interval were CO₂ levels higher than today. Interpretation of older $p$CO₂ has been hampered by discrepancies during some time intervals between two of the main ocean-based proxy methods used to reconstruct $p$CO₂: the carbon isotope fractionation that occurs during photosynthesis as recorded by haptophyte biomarkers (alkenones) and the boron isotope composition ($\delta^{11}$B) of foraminifer shells. Here we present alkenone and $\delta^{11}$B-based $p$CO₂ reconstructions generated from the same samples from the Plio~~cene and across a~~ -Pleistocene ~~glacial-interglacial cycle~~ at ODP Site 999~~. across a glacial-interglacial cycle.~~ We find a muted response to $p$CO₂ in the alkenone record compared to contemporaneous ice core and $\delta^{11}$B records, suggesting caution in the interpretation of alkenone-based records at low $p$CO₂ levels. This is possibly caused by the physiology of CO₂ uptake in the haptophytes. Our new understanding resolves some of the inconsistencies between the proxies and highlights that caution may be required when interpreting alkenone-based reconstructions of $p$CO₂.

## 1. **Introduction**

Understanding the absolute level and evolution of atmospheric $p\text{CO}_2$ through geological time is essential to our understanding of the Earth's Climate System. As both a fundamental, first order control, and a contributor to multiple dynamic feedbacks, atmospheric $p\text{CO}_2$ is critical in setting Earth's surface temperature (Lacis et al., 2010). Reconstructing $p\text{CO}_2$ evolution improves the understanding of both the mechanisms behind past climate change (Chalk et al., 2017) , and provides novel constraints on climate sensitivities (Martínez-Botí et al., 2015; PALAEOSENS, 2012). This then allows ground-truthing of our understanding the climate and the Earth system models that are used for predicting future climate change.

Over the past two decades, two common marine-based $\text{CO}_2$ proxies have emerged – alkenone-based $\varepsilon_p$ values ($\text{CO}_{2(\varepsilon p\text{-alk})}$), utilising the carbon isotopic fractionation imparted during photosynthesis in a subgroup of haptophytes (Bidigare et al., 1997), and planktic foraminiferal $\delta^{11}\text{B}$ values ($\text{CO}_{2(\delta 11\text{Bplank})}$), based on the pH control of boron speciation and isotopic fractionation in seawater (Hemming and Hanson, 1992). Multiple records of atmospheric $p\text{CO}_2$ now exist for the Cenozoic from both methods, showing a broadly similar long-term trend from a high-$\text{CO}_2$ greenhouse world of the early Cenozoic when $\text{CO}_2$ exceeded 400 µatm and may have been as higher than 1000 µatm to a low-$\text{CO}_2$ bi-polar glaciated world of the ~~present~~ late Pleistocene, when $\text{CO}_2$ fell to below 300 µatm (Anagnostou et al., 2016; Foster et al., 2017; Pagani et al., 2005, 2011; Pearson et al., 2009; Sosdian et al., 2018; Super et al., 2018).

However, discrepancies have recently become apparent between both methods when applied to the last 20 Ma (Badger et al., 2013a, 2013b). Specifically, the $\text{CO}_{2(\varepsilon p\text{-alk})}$ reconstructions often suggest a lower magnitude of short-term $p\text{CO}_2$ change compared to that from $\text{CO}_{2(\delta 11\text{Bplank})}$ (Badger et al., 2013a). Whilst this could be partially explained by mismatches between the sampling intervals, or by the influence of local surface water disequilibrium with the atmosphere with respect to $\text{CO}_2$, this discrepancy remains even for records generated from exactly the same sediment samples (Badger et al., 2013b vs Martínez-Botí et al., 2015). Both the $\text{CO}_{2(\varepsilon p\text{-alk})}$ and $\text{CO}_{2(\delta 11\text{Bplank})}$ have been used to estimate Earth system sensitivity in the Pliocene ~~(e.g. Pagani et al., (2009) vs. Martínez-Botí et al., (2015))~~ with differing results: $\text{CO}_{2(\varepsilon p\text{-alk})}$ suggesting higher than present earth system sensitivity Pliocene (7-10 °C per $\text{CO}_2$ doubling; Pagani et al., 2009), whilst $\text{CO}_{2(\delta 11\text{Bplank})}$ records sensitivity in line with our estimates for today (<5 °C per $\text{CO}_2$ doubling; Martínez-Botí et al., 2015). ~~; a~~Although this is at least partly due to the

different approaches used to calculate Earth system sensitivity in the two studies, it is also due to the differences in reconstructed $p\mathrm{CO_2}$ from the two approaches.

The $\mathrm{CO_{2(\epsilon p\text{-}alk)}}$ and $\mathrm{CO_{2(\delta 11Bplank)}}$ palaeobarometers are both based on mechanistic frameworks that have been calibrated in either the modern ocean or laboratory culture (Bidigare et al., 1997; Hemming and Hanson, 1992; Pagani et al., 2002; Sanyal and

Hemming, 1996). These proxies can be further ground-truthed in the recent geological past, when ice core records provide high-quality $p\mathrm{CO_2}$ data for the last 800 kyrs (Bereiter et al., 2015 and Table 1). In previous work, both $\mathrm{CO_{2(\delta 11Bplank)}}$ (Chalk et al., 2017; Foster, 2008; Foster and Sexton, 2014; Henehan et al., 2013; Hönisch and Hemming, 2005; Sanyal et al., 1995) and $\mathrm{CO_{2(\epsilon p\text{-}alk)}}$ (Jasper and Hayes, 1990) yield $p\mathrm{CO_2}$ records similar in absolute value and amplitude of change to those derived from ice cores. However, the emerging discrepancies between the two methods (Badger et al., 2013a, 2013b; Martínez-Botí et al.,

2015b) necessitate revisiting this validation, both between the two proxies, and between marine proxy and ice core reconstructions.

The ice core records $p\mathrm{CO_2}$ of the Pleistocene glacial-interglacial cycles (Bereiter et al., 2015 and Table 1; Petit et al., 1999) provide an opportunity for cross-calibrating proxy methods for determining atmospheric $p\mathrm{CO_2}$ in the geological archive ($\mathrm{CO_{2(\epsilon p\text{-}alk)}}$ and $\mathrm{CO_{2(\delta 11Bplank)}}$) with the direct-$\mathrm{CO_2}$ measurements from the ice cores.

**1.2 Study Site**

Ocean Drilling Program Site 999 is located in the Caribbean Sea (12° 44.639' N, 78° 44.360' W, 2838m water depth; Figure 1), has an orbitally calibrated age model and has been used previously for $\mathrm{CO_2}$ reconstructions. Our temporal sampling resolution is ~6 kyrs in the Pleistocene and ~9 kyrs in the Pliocene. Although $\mathrm{CO_{2(\epsilon p\text{-}alk)}}$ and $\mathrm{CO_{2(\delta 11Bplank)}}$ are independent of one another in many respects, they both rely on assumptions about the equilibrium of surface seawater with the atmosphere

with respect to $\mathrm{CO_2}$, sea surface temperature, and on well-constrained age models, which can make direct comparison between records from different sites difficult. Here we overcome these problems by producing $\mathrm{CO_{2(\epsilon p\text{-}alk)}}$ and $\mathrm{CO_{2(\delta 11Bplank)}}$ records from identical horizons in the same deep ocean sediment core in 1) the late Pleistocene, permitting direct comparison to ice core data (Figure 2a, Figure 3), and 2) across the intensification of Northern Hemisphere glaciation (INHG) in the Pliocene (Martínez-Botí et al., 2015a; Seki et al., 2010) (Figure 2b).

In terms of $CO_2$, ODP Site 999 in the Caribbean Sea is today slightly out of equilibrium with the atmosphere, with surface waters a little oversaturated in $CO_2$, providing a small net source of $CO_2$ to the atmosphere (~21 µatm; Takahashi et al., 2009). However the site has been shown to be suitable for recording past changes in $p$CO$_2$ (Foster, 2008; Foster and Sexton, 2014) and the air-sea equilibrium is not thought to have changed significantly from the Pliocene to today (see discussion in Bartoli et al., 2011). It is one of few sites where both alkenone and boron isotope records can be acquired given the good preservation of both foraminifera and organic matter (Badger et al., 2013b; Foster, 2008; Foster and Sexton, 2014; Martínez-Botí et al., 2015a), and Pliocene records of both are available (Badger et al., 2013b; Bartoli et al., 2011; Martínez-Botí et al., 2015a). It also has been demonstrated previously to record glacial-interglacial cycles of pH/CO$_2$ (Foster, 2008; Henehan et al., 2013) and a Pleistocene $CO_{2(\delta11Bplank)}$ record from 0-250 ka has been recently published (Chalk et al., 2017).

## 2 Methods

### 2.1 Alkenone isotopes

Our new alkenone based $CO_2$ record was calculated following Badger et al., (2013b), with modern day phosphate used in the estimation of the 'b' term, $U_{37}^{K'}$ ~~UK37~~' temperatures, and modern day salinity (35 psu). Samples were freeze dried, ground to a fine powder by hand, and extracted by Soxhlet apparatus using a dichloromethane (DCM) / methanol azeotrope (2:1, $v/v$) refluxing for 24 hours. Total lipid extracts were divided into three fractions (F) by small (4 cm) silica chromatography columns, with fractions eluting in 3 mL of $n$-hexane (F1), DCM (F2) and ethyl acetate/$n$-hexane (1:3. $v$:$v$, F3) respectively. Alkenones eluted in F2. Alkenone identification was confirmed by GC mass spectrometry (ThermoQuest Trace MS, He carrier gas) Alkenone isotope analyses were performed using a ThemoFisher Delta V connected via a Gas Chromatograph (GC) isolink and ConFlo IV to a Trace GC. The GC oven was programmed to increase in temperature from 70 °C to 200 °C at 20 °C min$^{-1}$, then to 300 °C at 6 °C min$^{-1}$ and held isothemal for 25 min. Conversion to VPDB was performed by reference to a laboratory standard gas of known $\delta^{13}$C and system performance was monitored using in-house fatty acid methyl ester and $n$-alkane standard mixtures of known isotopic composition. Long term precision is approximately 0.3 ‰. To estimate SST, the F2 fraction was also analysed by GC- flame ionisation detection (Hewlett Packard 5890 Series II), the GC oven was programmed

to increase in temperature from 70 °C to 130 °C at 20 °C min$^{-1}$, then to 300 °C at 4 °C min$^{-1}$ and held isothermal for 25 min. An approximately 50 m, 0.32 mm internal diameter capillary column with a 0.12 µm thick dimethylpolysiloxane equivalent film. A H$_2$ carrier gas was used, and quantification was monitored using an hexandecan-2-ol standard added prior to column chromatography. System performance as monitored with an in-house fatty acid methy ester standard. Alkenone ratios were

converted to SST using the global core-top calibration of (Müller et al., (1998), although this is a linear calibration, our uncertainty treatment (see below) should encompass any minor deviation from linear as $U_{37}^{K'}$ approaches 1 (see also the discussion in (Badger et al., (2013b). All alkenone analyses were carried out at the Bristol node of the of the NERC Life Sciences Mass Spectrometry Facility hosted by the Organic Geochemistry Unit, University of Bristol.

The alkenone isotope $\delta^{13}$C value is used to calculate the total carbon isotope fractionation that occurs during algal growth ($\varepsilon_p$).

This isotopic fractionation has been shown to be controlled by [CO$_2$]$_{(aq)}$ (equation 1; Jasper & Hayes 1990) which can then be converted to atmospheric CO$_2$ using Henry's law.

Equation 1.        $\varepsilon_p = \varepsilon_f - \dfrac{b}{[CO_2]_{(aq)}}$

To calculate $\varepsilon_p$ from alkenone $\delta^{13}$C vales the carbon isotopic composition of DIC is required, this is calculated from planktic foraminiferal calcite $\delta^{13}$C, whilst the fractionation which occurs during carbon fixation ($\varepsilon_f$), is here assumed constant. The 'b'

term is the sum of other physiological factors (such as growth rate, and cell size, and light limitation) which is estimated from the relationship shown in the modern ocean between 'b' and dissolved reactive phosphate [$PO_4^{3-}$] . Further details of the treatment can is detailed in Badger et al., (2013b).

Error bars in relevant figures are all 1sd and based on a full Monte Carlo propagation (n=10000) of the following uncertainties: ±2 °C and ±0.1 ‰ were applied to temperature and foraminiferal calcite $\delta^{13}$C, (normal probability function (pdf), 2σ error)

and ±2 and ±0.1 to salinity and [PO$_4^{3-}$], respectively (2σ; uniform pdf). Uncertainties on alkenone $\delta^{13}$C were estimated from replicate runs and calcite $\delta^{13}$C from repeat runs of an internal standard. Integrated analytical and calibration uncertainties for alkenone based temperatures were estimated and conservative estimates of likely variation for salinity and [PO$_4^{3-}$] were used. An 11 % error on the slope of b=a[PO$_4^{3-}$]+c was assumed, where a = 116.96 and c = 81.41 (Pagani et al., 1999).

For consistency with the $CO_{2(\delta 11Bplank)}$ record for this Site, we now adjust for the disequilibrium by subtracting the present day $CO_2$ surplus, and thus, have recalculated the included values of Badger et al., (2013b) accordingly. SSTs for our new Pliocene data were published in Davis et al., (2013).

## 2.2 Boron Isotopes

Boron isotope data were published in (Chalk et al., 2017) and are from the same core samples as our alkenone measurements. *Globigerinoides ruber sensu stricto* (white, n ~ 200 individuals from 300-355μm) samples were measured for boron isotope composition on Thermo Scientific Neptune MC-ICPMS at the University of Southampton according to methods described elsewhere (Foster et al., 2013; Martínez-Botí et al., 2015a; Rae et al., 2011). Analytical uncertainty is given by the external reproducibility of repeat analyses of Japanese Geological Survey Porites coral standard (JCP) at the University of Southampton

following Henehan et al., (2013) and is typically <0.2 ‰ (at 95 % confidence). Metal element:calcium ratios (Li, Mg, B, Na, Al, Mn, Ba, Sr, Cd, U, Nd, and Fe) were analyzed using an Thermo Element 2XR ICP-MS at the University of Southampton). Here, these data are used to assess adequacy of clay removal (Al/Ca <100 μmol/mol) and to generate down core temperature. pH and $CO_2$ were calculated using a Monte Carlo approach (uncertainties are 2sd, n = 10000 replicates) using R (R Core Team, 2015), for pH we use a boron isotopic composition of seawater of 39.6 ‰ (2sd of 0.1, Foster et al. 2010) and experimentally

determined isotopic fractionation factor (1.027, Klochko et al., 2006) as well as the species specific calibration for *G. ruber* of Henehan et al 2013 (also with incorporated uncertainties). For the $CO_2$ calculations we use a range of salinity (equal to used in the $CO_{2(\epsilon p-alk)}$ calculations) and total alkalinity (Talk) that encompasses the modern values (34-37 and 2100-2500 μM, respectively, both with a uniform rather than normal probability distribution. Temperature was determined using Mg/Ca of *G. ruber* following established methods (Delaney and Boyle, 1985; Evans and Müller, 2012)). Mg/Ca SST of planktic

foraminifera is used for $CO_{2(\delta 11Bplank)}$ and alkenone $U_{37}^{K'}$ for $CO_{2(\epsilon p-alk)}$ so that the carrier organisms for the $CO_2$ reconstruction and SST measurements match, ensuring the temperature measurement is coming from the appropriate part of the water column. Inorganic chemical constants were used from the seacarb package in R (Gattuso et al., 2015), and using published values for the pKB (Dickson, 1990). Reconstructed atmospheric $CO_2$ values from Foster, (2008) were recalculated to match this approach. All uncertainties are included in our simulation and are roughly equivalent to those assumed for the alkenone data

and are exactly the same as those used for Martinez-Boti et al. 2015 excluding the $\delta^{11}B_{sw}$, thus providing a fair comparison.

## 2.3 Coccolith length measurements

The uptake of $CO_2$ into the coccolithophore cell is effected by the cell size and geometry (Laws et al., 1997; Popp et al., 1998), using alkenones limits the variation of cell geometry by restricting the source organism to one with exclusively spherical cells (Laws et al., 1997; Popp et al., 1998), but some change in cell size is possible. Coccolith size is used as an semi-quantitative proxy for cell size because coccolith size is typically larger on larger cells, with that relationship being broadly consistent within a single taxonomic group where growth behaviour is broadly comparable (Gibbs et al., 2013; Henderiks, 2008; Sheward et al., 2017). Long-axis coccolith length measurements were therefore taken from 100 specimens of the family Noelaerhabdaceae per sample from standard smear slides. Specimens were imaged at 1500x magnification and measured using CellD software.

To investigate the potential influence of changing cell size on $CO_{2(\varepsilon p\text{-}alk)}$ Equation 1 can be adapted:

Equation 2
$$\varepsilon_p = \varepsilon_f - \frac{b\prime}{[CO_2]_{(aq)}}$$

With b' calculated from using the volume to surface area ratio (V:SA) of modern and fossil coccospheres (Equation 3; Henderiks and Pagani, 2007)

Equation 3
$$b' = b\frac{V:SA_{fossil}}{V:SA_{Ehux}}$$

$V:SA_{Ehux}$ is 0.9 ±0.1 µm in modern haptophytes (Popp et al., 1998) and $V:SA_{fossil}$ can be estimated from lith size measurements (Equation 4; Henderiks and Pagani, 2007)

Equation 4
$$D_{cell} = 0.55 + 0.88L_{coccolith}$$

**2.4 Age Model**

For the interval 0-500 ka, we generated a detailed age model by tuning the planktic foraminifer (*G. ruber*) $\delta^{18}O$ record from Site 999 (at ~0.5 to 2.0 kyr resolution) (Schmidt et al., 2006) to the LR04 benthic $\delta^{18}O$ stack (Lisiecki and Raymo, 2005) using the Analyseries software (Paillard et al., 1996), the Pliocene portion of Site 999 is part the LR04 stack and that astronomically tuned age model is used here (Lisiecki and Raymo, 2005).

**2.5 Bayesian exploration of CO$_{2(\varepsilon p\text{-alk})}$ input variables**

In order to examine the influence of the various input parameters for the calculation of $p$CO$_2$ from alkenone $\delta^{13}C$ values, we carry out a second set of Monte Carlo simulations (n=100,000) with expanded uncertainty. In this case, we more fully explore uncertainty space using the following input uncertainties (at 95 % confidence or full range): SST (normal distribution, $\pm$ 6 °C), $\varepsilon_f$ (uniform distribution, 24 to 28), b (normal distribution, $\pm$ 40), CO$_2$ disequilibrium (20 $\pm$ 20). These input distributions are our *prior* distributions. We then evaluate the CO$_2$ output for each alkenone sample against synchronous ice core $p$CO$_2$ and boron isotope $p$CO$_2$ for the Pleistocene and Pliocene, respectively. By only selecting those simulated alkenone $p$CO$_2$ levels that agree with ice core or $p$CO$_{2(\delta11Bplank)}$ (including associated uncertainties), we can re-evaluate the input distributions (our *posterior*) and gain insights into the relative importance of each of the input variables in potentially driving the observed disagreements in $p$CO$_2$. Uncertainties in the $p$CO$_{2(\delta11Bplank)}$ are as described above, and we apply an uncertainty of $\pm$ 6 ppm (2s) for the ice core $p$CO$_2$ record (Ahn et al., 2012).

**3 Results and Discussion**

Alkenone and *G. ruber* $\delta^{13}C$ values (Figure 3a,e) were used to calculate $\varepsilon_p$ values (Figure 3c,g). Alkenone $\delta^{13}C$ values are relatively stable through the Pleistocene portion of the record, varying between -24.5 ‰ and -23.2 ‰. Values are slightly higher in the Pliocene, varying between -24.1 ‰ and -21.7 ‰, *G. ruber* $\delta^{13}C$ values are relatively stable through the whole record, varying between 0.53 ‰ and 1.57 ‰. These give rise to $\varepsilon_p$ values which are similarly fairly stable, varying between 10.5 ‰ and 12.2 ‰ in the Pleistocene, and between 9.53 ‰ and 11.8 ‰ in the Pliocene. Our $U^{K'}_{37}$ ~~UK37'~~ SST (Figure 3c,g) record shows warmer temperatures in the Pliocene of around 27 °C, with cooler temperatures recorded in the Pleistocene, with the coldest SST recorded in the glacial which is ~2 °C cooler than the interglacial. These record are combined (see Methods)

to produce the $p$CO$_2$ record (Figure 2, 3c,f) which shows largely stable and invariant values through both the Pliocene and Pleistocene portions of our record. ~~We estimate the 'b' term of equation 1 using the modern day relationship observed between 'b' and [PO$_4^{3-}$]. This term combines all other physiological factors which may influence $\varepsilon_p$ including cell size, growth rate and light limitation.~~

Published low temporal resolution Pliocene records from Site 999 (Seki et al., 2010), using both the CO$_{2(\varepsilon p\text{-alk})}$ and CO$_{2(\delta 11Bplank)}$ palaeobarometers, show a $p$CO$_2$ decrease at ~ 2.8 Ma. However, this agreement relies on correcting the CO$_{2(\varepsilon p\text{-alk})}$ for changes in haptophyte cell size, which was based on a low temporal resolution lith size record (Seki et al., 2010). Changes in haptophyte cell size alter the volume:surface area ratio available for gaseous exchange, and can therefore modify the fractionation recorded by CO$_{2(\varepsilon p\text{-alk})}$(Popp et al., 1998). Our new CO$_{2(\varepsilon p\text{-alk})}$ record  at Site 999 now spans 3.3- 2.6 Ma at higher temporal resolution,

supplementing data from Badger et al.,_(2013b). A lith size record has also been generated for the same samples used for CO$_{2(\varepsilon p\text{-alk})}$ for 3.3-2.6 Ma (Davis et al., 2013). We find no evidence to support the change in lith size applied by Seki et al., (2010) with lith size (and hence cell size) remaining stable across the primary $p$CO$_2$ change at 2.8 Ma (Davis et al., 2013). Consequently, although our new CO$_{2(\varepsilon p\text{-alk})}$ record is higher resolution than that of Seki et al., (2010), we no longer have any evidence for the cell size shift at 2.7 Ma (Figure 4).

We compare our record with the CO$_{2(\delta 11Bplank)}$ records of Martínez-Botí et al., (2015) in Figure 2. With the cell size correction now removed, the decrease in CO$_{2(\varepsilon p\text{-alk})}$ across the INHG, and the agreement of CO$_{2(\varepsilon p\text{-alk})}$ with CO$_{2(\delta 11Bplank)}$, both now disappear (red symbols, Figure 2b). As such, CO$_{2(\varepsilon p\text{-alk})}$ for the whole of this Pliocene interval (2.6 – 3.3 Ma) remains stable and low (mean CO$_{2\,(\varepsilon p\text{-alk})}$ =251±13; 1σ min=228 max=286 $\mu$atm), whereas CO$_{2(\delta 11Bplank)}$ is on average higher and more variable (mean CO2$_{(\delta 11Bplank)}$=342±50; 1σ min=234 max=452 $\mu$atm).

In the Pleistocene, our CO$_{2(\varepsilon p\text{-alk})}$ record covers one complete glacial-interglacial (G-IG) cycle from 110 – 260 ka, encompassing Marine Isotope Stage (MIS) 5,6,7 the end of MIS 8, and terminations II and III (red open diamonds, Figure 2a).  The CO$_{2(\delta 11Bplank)}$ record of Chalk et al. (2017) covers two G-IG cycles from the late Holocene to MIS8 (blue open circles, Figure 2a). $\delta^{11}$B$_{plank}$ closely tracks the rise and fall of $p$CO$_2$ derived from ice cores (Chalk et al., 2017), with CO$_{2(\delta 11Bplank)}$ exhibiting similar values to atmospheric CO$_2$ within uncertainty (Figure 2a), and with only small deviations from ice core CO$_2$ as a result

of: (*i*) the noise in the reconstruction; and (*ii*) perhaps a small diagenetic effect on $CO_{2(\delta11Bplank)}$ relating to periods of carbonate dissolution in portions of the core which show high foraminiferal fragmentation (e.g. MIS 5d; Schmidt et al., 2006).

In contrast, $CO_{2(\varepsilon p-alk)}$ is within error of the ice core data only during the interglacials when $CO_2$ partial pressures are similar to those of the pre-industrial era. Crucially, $CO_{2(\varepsilon p-alk)}$ clearly fails to record the lower $pCO_2$ of the glacials, remaining at around

260 $\mu$atm throughout (mean $CO_{2(\varepsilon p-alk)}$=259±27; Figure 2a). This concentration of $pCO_2$ is also very close to that recorded by $CO_{2(\varepsilon p-alk)}$ in the Pliocene at this Site (mean $CO_{2(\varepsilon p-alk)}$=252±26 $\mu$atm; Figure 2). Similar alkenone behaviour has also been observed in another, albeit lower resolution, record from ODP Site 925 (Zhang et al., 2013; Figure 2), where the $CO_{2(\varepsilon p-alk)}$ remains unchanged during the Pleistocene (20 – 170 Ka) and Pliocene.

Overall, these results suggest that, at least at these sites, the $CO_{2(\delta11Bplank)}$ palaeobarometer does faithfully record atmospheric

$CO_2$ change, whereas the $CO_{2(\varepsilon p-alk)}$ proxy is unable to reconstruct the low levels of atmospheric $CO_2$ during the glacial. This suggests that, in its present and frequently applied form, $CO_{2(\varepsilon p-alk)}$ is not accurately recording atmospheric $CO_2$, and this could explain the discrepancy between the Pliocene $CO_{2(\varepsilon p-alk)}$ and $CO_{2(\delta11Bplank)}$ records. We further evaluate this by using regression analysis between ice core and the paired-proxy data (Figure 5). $CO_{2(\delta11Bplank)}$ levels are largely consistent with those determined from ice cores, clustering around the 1:1 line with a slope also close to 1 (0.95±0.13) (Figure 5a), whereas variance in $CO_{2(\varepsilon p-}$

$_{alk)}$ is strongly muted compared to that observed in the ice core data (Figure 5b).

As both cell size (Popp et al., 1998) and growth rate (Bidigare et al., 1997) can modify $\delta^{13}C_{alk}$ via the 'b' term, we investigated whether either of these could explain the muted response of $CO_{2(\varepsilon p-alk)}$ to atmospheric $CO_2$. Hapytophyte cell size can be estimated from their lith size, but as noted above, there is no evidence for significant changes in the Pliocene (Davis et al., 2013) nor is there evidence for any change across MIS5-8 (Figure 4a). There is an overall reduction in mean lith size from the

Pliocene to the Pleistocene (Figure 4a, b), which could offset a long term $pCO_2$ decline and thus explain the apparent lack of difference between Pliocene and Pleistocene $CO_{2(\varepsilon p-alk)}$ at Site 999 (Figure 6). However, this longer-term reduction in lith size cannot explain the muted response to Pleistocene G-IG $CO_2$ change.

Growth rate is more difficult to reconstruct; most available proxy systems reconstruct phytoplankton or whole ecosystem productivity, rather than coccolithophorid growth rate. However emerging trace metal datasets do suggest changing productivity on glacial-interglacial timescales at Site 999, with lower productivity in the glacial (Trumbo, 2015). If lower productivity is linked with a simultaneous reduction in growth rate, then it could explain some of the lack of signal in $CO_{2(\varepsilon p\text{-}alk)}$, however to reduce the $CO_{2(\varepsilon p\text{-}alk)}$ sufficiently to overlap with ice core $CO_2$ would require an order of magnitude reduction in growth rates during the glacial. ~~.~~

This suggests that either our understanding of growth rate effects on $CO_{2(\varepsilon p\text{-}alk)}$ is incorrect, or the estimation of cell size using preserved liths does not capture original cell size variations, or a combination of these or other factors leads to the rather muted trends in $CO_{2(\varepsilon p\text{-}alk)}$ through the glacial-interglacial cycle.

The failure of $CO_{2(\varepsilon p\text{-}alk)}$ from these two sites to record the G-IG $pCO_2$ variation also necessitates reassessment of earlier $CO_{2(\varepsilon p\text{-}alk)}$ studies that were able to reconstruct such changes. For instance whilst Jasper and Hayes, (1990) replicated the $CO_2$ change over the last 100 kyrs of the Vostok ice core from DSDP Site 619 (Gulf of Mexico)~~; Figure 1,6~~) and ~~(~~Bae et al., (2015) are able to replicate the ice core data at a site in the Japan Sea, records from the Arabian Sea (Palmer et al., 2010), Angola Current (Andersen et al., 1999) and the equatorial Atlantic (Zhang et al., 2013) either fail to record ice core $CO_2$ or require additional corrections to do so (Figure 7). ~~. However,~~ It may also be that at some sites, such as at ~~a record from the~~ Equatorial Pacific Site (MANOP site C~~;~~ (Figure 1, 7~~6~~)~~-~~ these records represent changing air-sea disequilibrum ~~also failed to replicate the *pCO₂* changes observed in the ice core data over the last 255 kyrs, and instead was interpreted as recording changes in air-sea equilibrium, not atmospheric CO₂~~ (Jasper et al., 1994). ~~Smoothing and correction for estimated growth rates revealed the gross features of the ice core record (Stoll and Schrag, 2000), but still only recorded 30-35 % of the variance in the ice core data (Bereiter et al., 2015; Stoll and Schrag, 2000). These~~ T~~t~~wo of these studies interpreted their data using a different $\varepsilon_p$ relationship than later work; when these data are recalculated using the more recent model the patterns remain unchanged (Figure 7~~6~~). What is more, in a global alkenone $\delta^{13}C$ calibration study (Pagani et al., 2002) aimed at replicating Holocene atmospheric conditions it was noted that low latitude (sub-tropical) sites perform poorly, consistent with our observations. Considering this

present study and previously published work, 3 5 out of 4 7 late Pleistocene alkenone $\delta^{13}C$ studies do not show the variations in $pCO_2$ evident from contemporaneous ice core records (Figure 2; Figure 7 6).

Our Bayesian approach allows us to explore the $CO_{2(\epsilon p\text{-}alk)}$ proxy, as it was mathematically expressed by Bidigare et al (1997) and subsequent authors, and test which what variables may be responsible for causing the observed disagreements with the ice

core and $CO_{2(\delta 11Bplank)}$ records given a largely invariant $\epsilon_p$ for the Pleistocene. Figure 8 7 illustrates the prior distributions of the input variables (blue) and an example posterior for the alkenone sample at 150 kyr (red). As can be seen in this example, selecting only those simulations of $CO_{2(\epsilon p\text{-}alk)}$ that overlap with the ice core $CO_2$ for this time interval shifts the distributions such that an agreement is found when b is lower than the prior, $\epsilon_f$ tends to be higher than the prior and SST and $CO_2$ disequilibrium are little different. Figure 9 8 shows the posterior median and 95 % distribution of b, $\epsilon_f$ and SST for all the

samples from the Pleistocene and Pliocene in time series. Patterns that emerge are illustrated in Figure 10 9, where a negative relationship between $pCO_2$ and posterior $\epsilon_f$ and a positive relationship between $pCO_2$ and posterior b and SST is evident. For SST it should also be noted that for the Pleistocene the posterior correlates well with the prior, while for the Pliocene it is significantly elevated (Figure 10 9), perhaps suggesting a role for incorrect SST in driving some of the lack of Pliocene to Pleistocene change in $CO_{2(\epsilon p\text{-}alk)}$ observed (Figure 2). This SST change would however need to be substantial and go beyond

the ±2 °C we include in our uncertainty propagation, and would also potentially influence $CO_{2(\delta 11Bplank)}$, further complicating this finding.

We recognize that the nature of the patterns we observe here is a function somewhat of the range used for each input term. The chosen ranges are however conservative, but realistic, assessments of the likely uncertainty associated with each term. For instance, b ± 40 encompasses the residual scatter around the relationship between b and [PO₄] described by Pagani et al.,

(2005). In addition to pointing towards a potential underestimate of Pliocene SST with the $U_{37}^{K'}$ $U^K_{37}$ proxy at ODP 999, this Bayesian treatment supports the assertion that the current understanding of the $CO_{2(\epsilon p\text{-}alk)}$ proxy is wanting and that the b term may not in fact capture the scaling of the relevent physiological parameters or those that are truly important. In particular, it appears that the physiological parameters packaged in the b-term, and potentially the degree of fractionation upon fixation, $\epsilon_f$, are themselves a function of $CO_2$ or some parameter that correlates with $CO_2$ (e.g. temperature, nutrients, growth rate etc.).

As noted above mean lith size is significantly different for the Pliocene and Pleistocene. A comparison of our posterior b and lith size does reveal a good correlation between these variables (Figure 11; $r^2 = 0.52$, p<<0.01), though this is largely, but not exclusively, a function of the mean change across the Plio-Pleistocene. Importantly, the observed relationship between b and lith size is very different from that described in (Henderiks and Pagani, (2007). Suggesting that if lith size is important, our understanding, at least as laid out in (Henderiks and Pagani, (2007) is incorrect.

An alternative explanation however could be that the invariant parameterisation of physiological factors into the 'b' term model could beis simply flawed in general, or is at least lacking important components. The dominant species producing alkenones in this part of the Caribbean today, and likely since its first appearance 268 kyrs ago, is *Emiliania huxleyi* (Winter et al., 2002). *E. huxleyi* first appears 290 kyrs ago, but did not become the dominant Noelaerhabdaceae until ~82 Ka when it began to outcompete the closely related *Gephyrocapsa* spp. which in in turn took over from *Reticulofenestra* in the late Pliocene (Gradstein et al., 2012; Raffi et al., 2006). Both our Pleistocene alkenone recordss are therefore a composite of closely-related but distinct noelaerhabdaceaen species, with nether record dominated by *E. huxleyi*. We cannot rule out that there could be physiological differences between the extant *E. huxleyi* species and the alkenone producers for our record. However the Site 925 $CO_{2(\varepsilon p\text{-alk})}$ record of the last glacial-interglacial cycle, which would been primarily sourced from *E. huxleyi* is similarly flat, suggesting that species specific biosynthesis differences are unlikely to be the whole story. The *Reticulofenestra-Gephyrocapsa-Emiliania* lineage has strong stratigraphic, morphological and genetic support, with *Emiliani* and *Gephyrocapsa* only recently genetically diverging (Bendif et al., 2016). Likely these taxa shared the same or similar ecologies. Recent experimental work has shown that this globally important species has evolved a carbon concentrating mechanism (CCM) to respond to limiting $CO_2$ by upregulating genes at low DIC to maintain carbon requirements (Bach et al., 2013). CCMs result in a breakdown of the relationship between $\varepsilon_p$ and $CO_2$ as defined and calibrated by Bidigare et al., (1997). It has been thought that the increased expression of CCMs will cause $\varepsilon_p$ values to decrease, due to the isotopic offset between $CO_{2(aq)}$ and $HCO_3^-$ and decreased carbon leakage from the cell (Zhang et al., 2013), effectively exacerbating the expected trend towards lower $\varepsilon_p$ values at lower $pCO_2$ and inconsistent with our observation of relatively stable $\varepsilon_p$ values across G-IG cycles.

However, CCMs appear to modulate carbon flow across cellular compartments (e.g. cytosol, chloroplast and calcification vesicle), and could also yield elevated rather than lower $\varepsilon_p$ due to the concentrating of $CO_2$ at the site of carbon fixation (Bolton and Stoll, 2013). Additionally, as temperature modulates resource allocation between biosynthesis and photosynthesis (Sett et al., 2014), $CO_2$ optima are species specific and vary with temperature which may explain why some sites in the region with different dominant haptophyte species are capable of recording G-IG changes, whilst others struggle. as temperature modulates resource allocation between biosynthesis and photosynthesis (Sett et al., 2014). Furthermore, it has been postulated that changes in carbonate chemistry affect the redox state inside *E. huxleyi* cells which subsequently causes a reorganization of carbon flux within and across cellular compartments (Rokitta et al., 2012). Such a re-distribution of inorganic carbon amongst different pathways also likely influences $\varepsilon_p$ and is currently not mechanistically represented by Bidigare et al., (1997) and other models.

## 4. Conclusions

Our data show that the classical application of the alkenone $p$CO$_2$ proxy fails to capture glacial-interglacial changes observed in the ice cores. With increased confidence in $CO_{2(\delta11Bplank)}$ supplied by that proxy's ability to capture Pleistocene $p$CO$_2$ variability, our data also suggest that the discrepancy between $CO_{2(\delta11Bplank)\text{-}}$ and $CO_{2(\varepsilon p\text{-}alk)}$ in the Pliocene may also be due to problems with $CO_{2(\varepsilon p\text{-}alk)}$. Emerging insights into coccolithophore $CO_2$ allocation pathways and their sensitivity to $CO_2$ and temperature, in conjunction with our inter-proxy comparisons, indicate that the long-standing $CO_{2(\varepsilon p\text{-}alk)}$ proxy requires major revision and recalibration. If CCMs are preferentially more important for the alkenone palaeobarometer than growth rate, the muted alkenone palaeobarometer response may be limited to the low $CO_2$ world of the Plio-Pleistocene and particularly in tropical waters where $CO_{2[aq]}$ is especially low. By extension, this proxy (and interpretations based on it) likely retains utility at the higher $CO_2$ levels typical of the early Cenozoic (and at high latitudes where $CO_{2[aq]}$ is high) where active carbon uptake is less likely (Zhang et al., 2013). This is especially true if haptophyte CCMs only evolved in the late Miocene as a response to declining $CO_2$ levels (Bolton and Stoll, 2013). Regardless, the discrepancy between $CO_{2(\varepsilon p\text{-}alk)}$ and ice core $CO_2$ records indicates that alkenone isotopes in several locations do not faithfully record atmospheric $CO_2$ at relatively low, Plio-Pleistocene-like $CO_2$ levels. Furthermore, the muted response of $CO_{2(\varepsilon p\text{-}alk)}$ to $[CO_{2(aq)}]$ at lower concentrations calls into

question the underlying basis of the high climate sensitivities previously reconstructed using this method in the Plio-Pleistocene (Pagani et al., 2009). This, coupled with further evidence of the fidelity of $CO_{2(\delta11Bplank)}$ at Site 999 suggests  that the climate sensitivities derived from $CO_{2(\delta11Bplank)}$ (which are consistent with climate models used both in palaeoclimate and future climate projections) are more accurate (Martínez-Botí et al., 2015a).

## Author contributions

MPSB and GLF conceived the study [conceptualization], MPSB, TBC and GLF designed the methodology, carried out data collection and analysed the data [formal analysis, investigation, methodology]. PRB, SJG, HP and AM performed data collection, PFS finalised the age model [investigation]. MPSB wrote the manuscript and prepared figures [Visualisation, Writing – original draft]. RDP (PI) and GLF and DNS (CoIs) supervised the project and acquired funding [Funding acquisition & Supervision]. All authors contributed to interpretation, writing and reviewing the manuscript [Writing – review & editing].

## Acknowledgements

This study used samples provided by the International Ocean Discovery Program (IODP). We thank Alex Hull and Gemma Bowler for laboratory work, Lisa Schönborn and Günter Meyer for technical assistance, Alison Kuhl and Ian Bull for research support, and Andy Milton at the University of Southampton for maintaining some of the mass spectrometers used in this study. This study was funded by NERC grant NE/H006273/1 to RDP, DNS and GLF (which supported MPSB). We also acknowledge the ERC Award T-GRES and a Royal Society Wolfson Research Merit Award to RDP. GLF is also supported by a Royal Society Wolfson Research Merit Award. We thank Kirsty Edgar for comments on an early draft of the manuscript, the two anonymous reviewers of this submission, and reviewers through various rounds of review whose comments greatly improved the manuscript. We are grateful to Thomas Bauska for encouraging us to do better at referencing the ice core data, and John Jasper for discussion of the early days of the alkenone palaeobarometer.

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

**Tables**

Table 1: Sources of ice core data used throughout as compiled by (Bereiter et al., (2015)

| Age interval (Kyr BP) | Ice core location | Reference |
|---|---|---|
| -0.051 – 1.8 | Law Dome | Rubino et al., (2013) |
| 1.8 – 2 | Law Dome | MacFarling Meure et al., (2006) |
| 2 – 11 | Dome C | Monnin et al., 2001, (2004) |
| 11 – 22 | WAIS | Marcott et al., (2014) |
| 22 – 40 | Siple Dome | Ahn and Brook, (2014) |
| 40 – 60 | TALDICE | Bereiter et al., (2012) |
| 60 – 115 | EDML | Bereiter et al., (2012) |
| 105 – 155 | Dome C Sublimation | Schneider et al., (2013) |
| 155 – 393 | Vostok | Petit et al., (1999) |

**Figures**

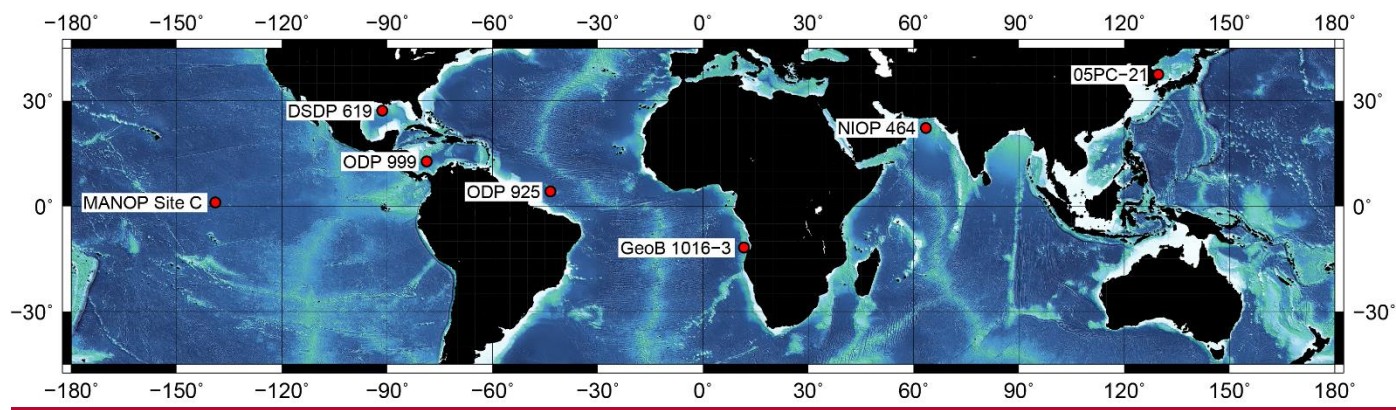

**Figure 1 Site map. Locations of sites discussed the text.**

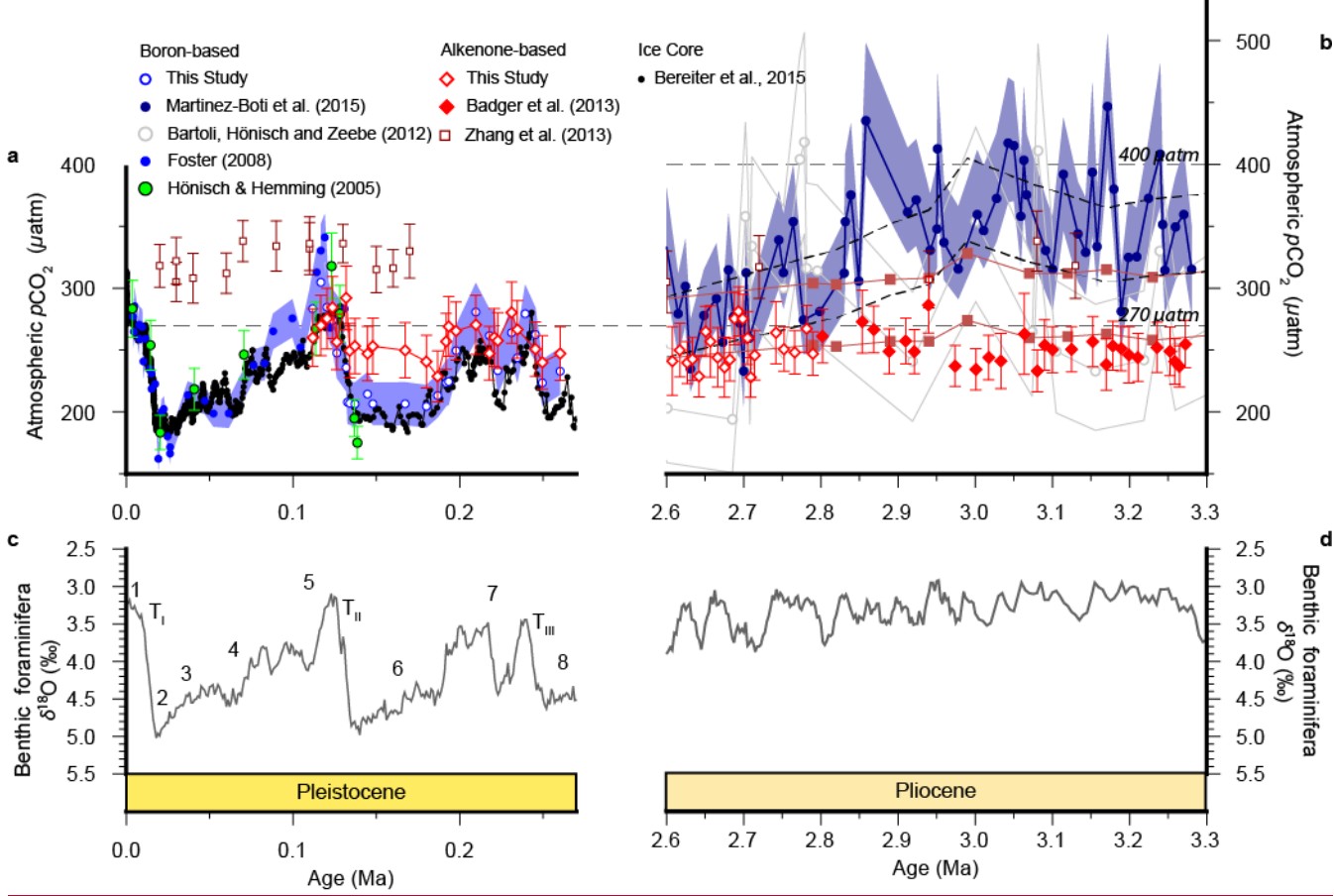

**Figure 2 Atmospheric CO₂ reconstructions through the Plio-Pleistocene. a: Published boron isotope CO$_{2(\delta11Bplank)}$ records from ODP Site 999 (open blue circles;** (Chalk et al., 2017)**, bright blue filled circles; (Foster, 2008 recalculated as described in the text), grey open circles** (Bartoli et al., 2011)**, dark blue filled circles** (Martínez-Botí et al., 2015)**) and DSDP Site 668 (green filled circles** (Hönisch and Hemming, 2005)**); b: published CO$_{2(\epsilon p\text{-}alk)}$ records from ODP Site 925 (maroon open squares** (Zhang et al., 2013)**) and ODP Site 999 (red filled diamonds**(Badger et al., 2013b) **and ice core records (black filled squares** (Bereiter et al., 2015 ~~and Table 1; Petit et al., 1999~~))**, as well as our new alkenone isotope records from ODP Site 999 (red open diamonds). The lith-size corrected (black dashed envelope) and uncorrected (red solid envelope) of Seki et al.,** (Seki et al., 2010) **are also shown. All records are shown with 1σ uncertainties as described elsewhere. c: benthic foraminiferal stable oxygen isotope stack** (Lisiecki and Raymo, 2005) **with Marine Isotope Stages (MIS; numerals) and Terminations (T) indicated.**

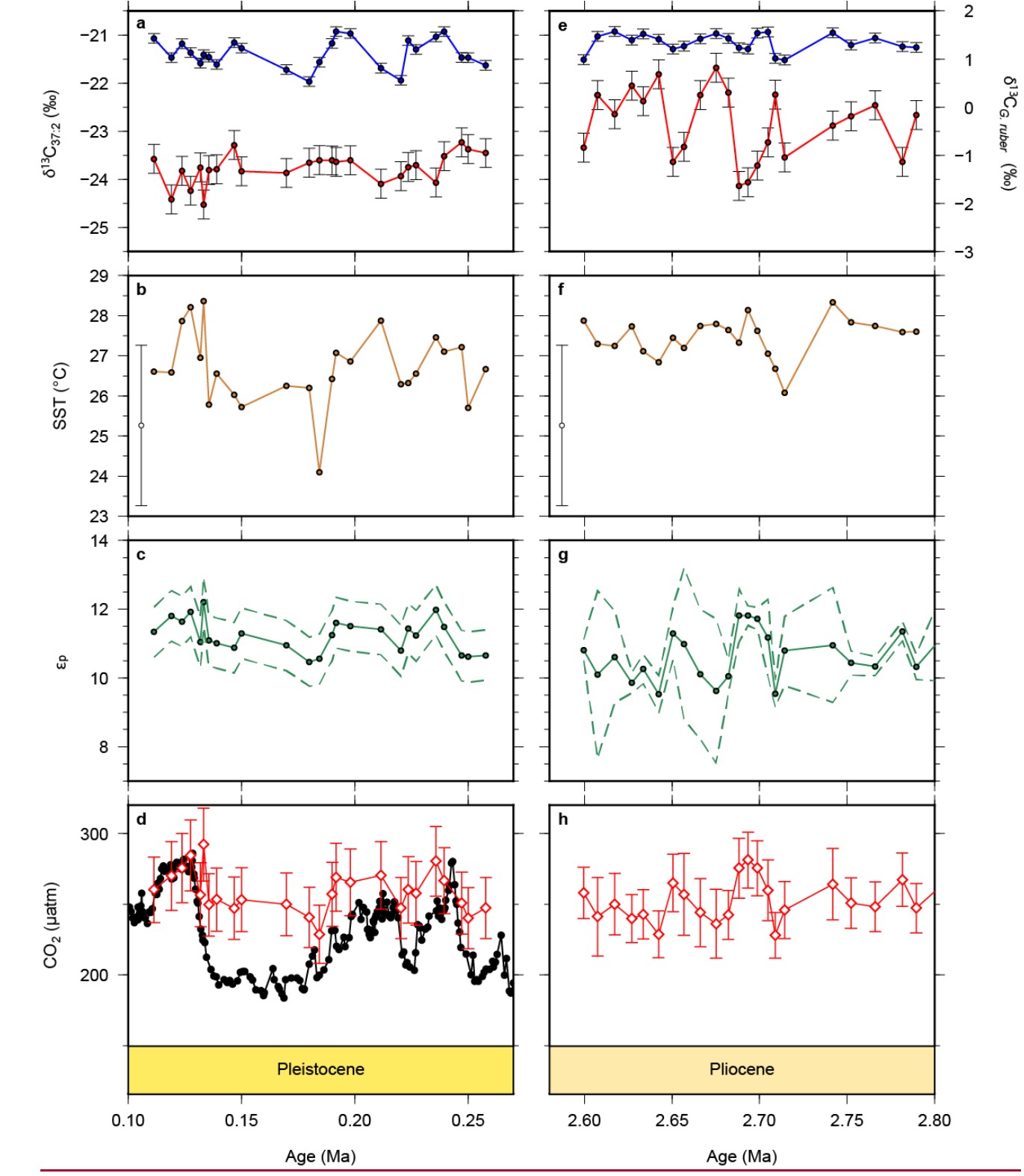

**Figure 3 New and recalculated date for CO$_{2(\varepsilon\text{p-alk})}$ for the Pleistocene and Pliocene from ODP Site 999. Alkenone $\delta^{13}$C values are shown as red circles for the Pleistocene (a) and Pliocene (b) with *G. ruber* $\delta^{13}$C from the same samples shown in blue. Alkenone unsaturation-derived SST is shown for the Pleistocene (b) and Pliocene (f). The Pliocene SST data has been previously published as Davis et al., 2013 and is from the same samples as our alkenone $\delta^{13}$C values. Calculated $\varepsilon_P$ data for the Pleistocene (c) and Pliocene (g) and atmospheric $p$CO$_2$ from CO$_{2(\varepsilon\text{p-alk})}$ for the Pleistocene (d) and Pliocene (h) (red diamonds). Ice core $p$CO$_2$ data is shown for the Pleistocene (black circles) for comparison (Bereiter et al., 2015 and Table 1).**

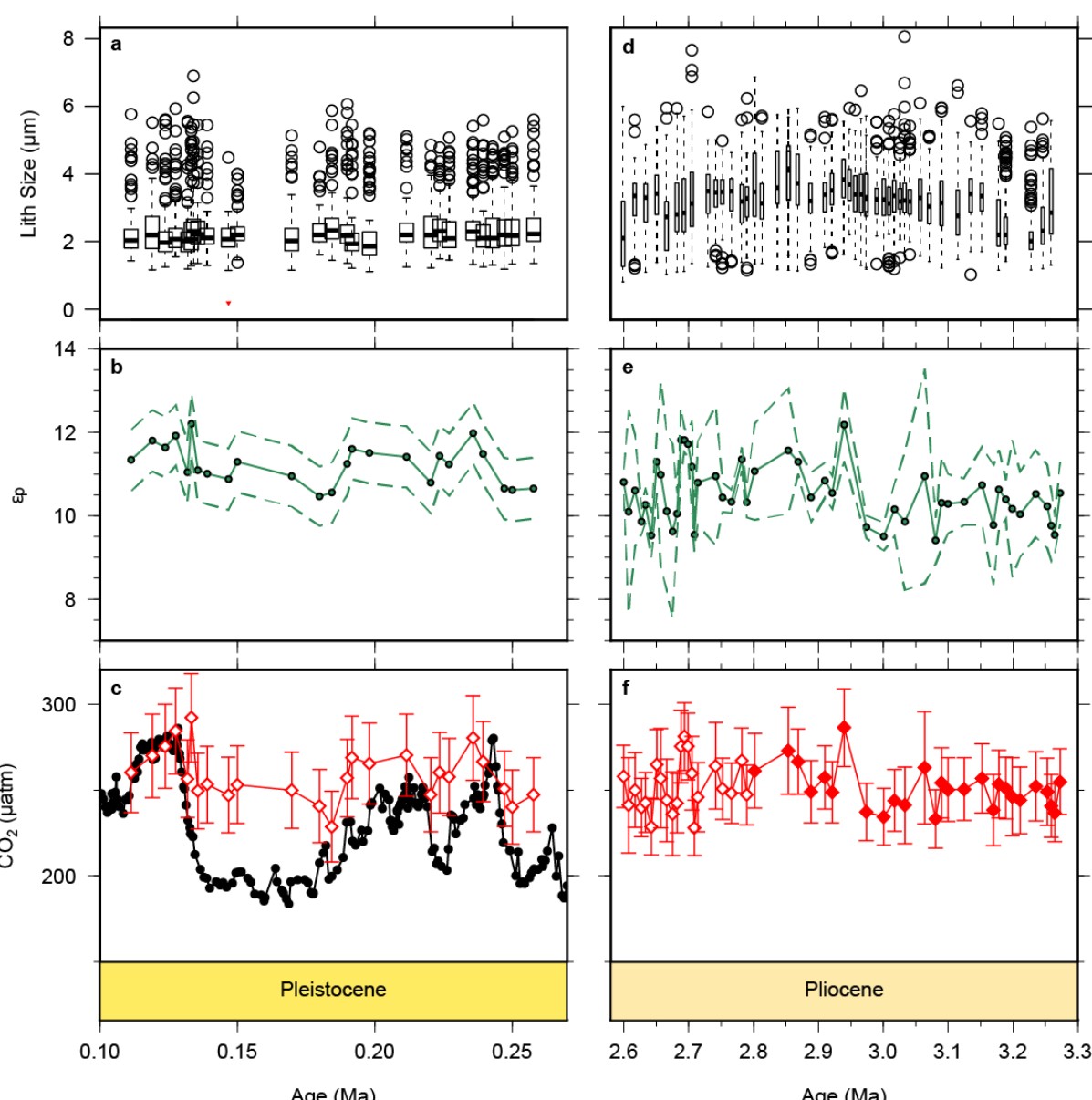

**Figure 4 Lith Size data for samples used for CO₂ calculations. Pleistocene lith size (a̶b̲) are from this study, whilst Pliocene (b) values were published previously** (Davis et al., 2013) **but are from the same samples as our CO₂ estimates. Pleistocene $\varepsilon_p$ (b) are from this study, whilst the Pliocene data is from this study (2.6-2.8 Mar) and for** Badger et al., (2013b) **( 2.8-3.3 Ma). The lower panels show CO₂($\varepsilon_{p\text{-alk}}$) for the Plesistocene (c) and Pliocene (f) as red diamonds. The filled diamonds in (f) are** (Badger et al., 2013a)**. The Pleistocene ice core data** (Bereiter et al., 2015 and Table 1) **are shown for comparison in (c). The drop in lith size from the Pliocene to Pleistocene is similar to what has been documented previously** (YOUNG, 1990)**. Outliers in a and d were calculated following the 1.5 rule in R (R Core Team, 2015).**

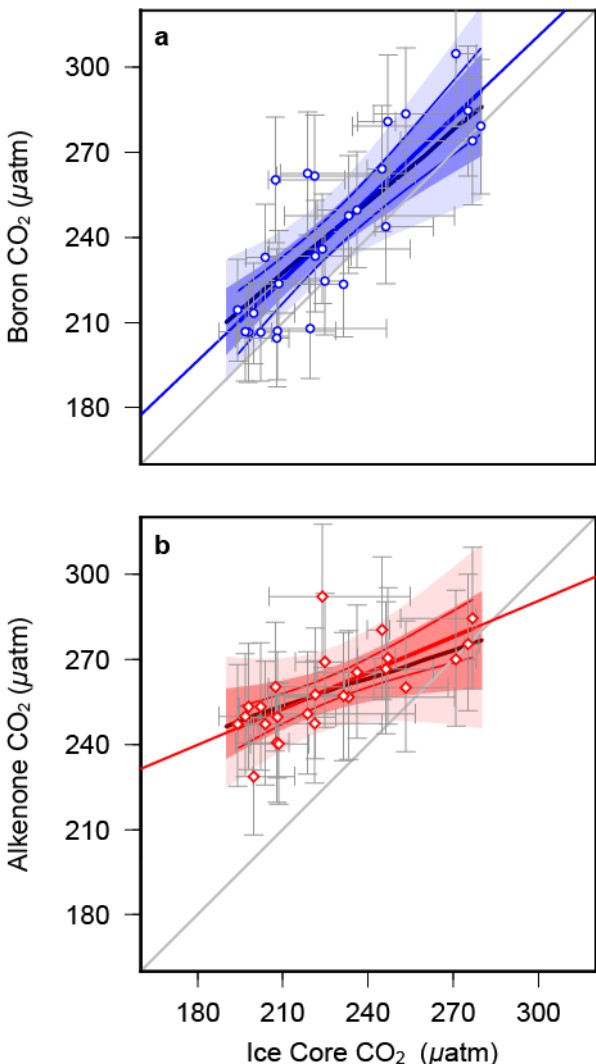

**Figure 5** Regression analyses of proxy-based $p$CO$_2$ with ice core data a; CO$_{2(\delta 11Bplank)}$ (Chalk et al., 2017) **and b; CO$_{2(\epsilon p\text{-alk})}$ vs ice core data** (Bereiter et al., 2015 and Table 1) **for MIS5-8, interpolated in the age domain. Regression lines (in red/blue) are linear fits with 68 and 95 % confidence intervals, calculated by bootstrapping the uncertainties in proxy $p$CO$_2$ (Monte Carlo method described in methods). Uncertainty in the ice core values are by estimated by applying a 3000 uncertainty in the age model during interpolation. Uncertainty envelopes considering data points alone (no bootstrap) are solid lines, with pmax regressions in the thicker, darker colours. A 1:1 line is shown in grey for comparison. Statistical calculations were performed in R (R Core Team, 2015).**

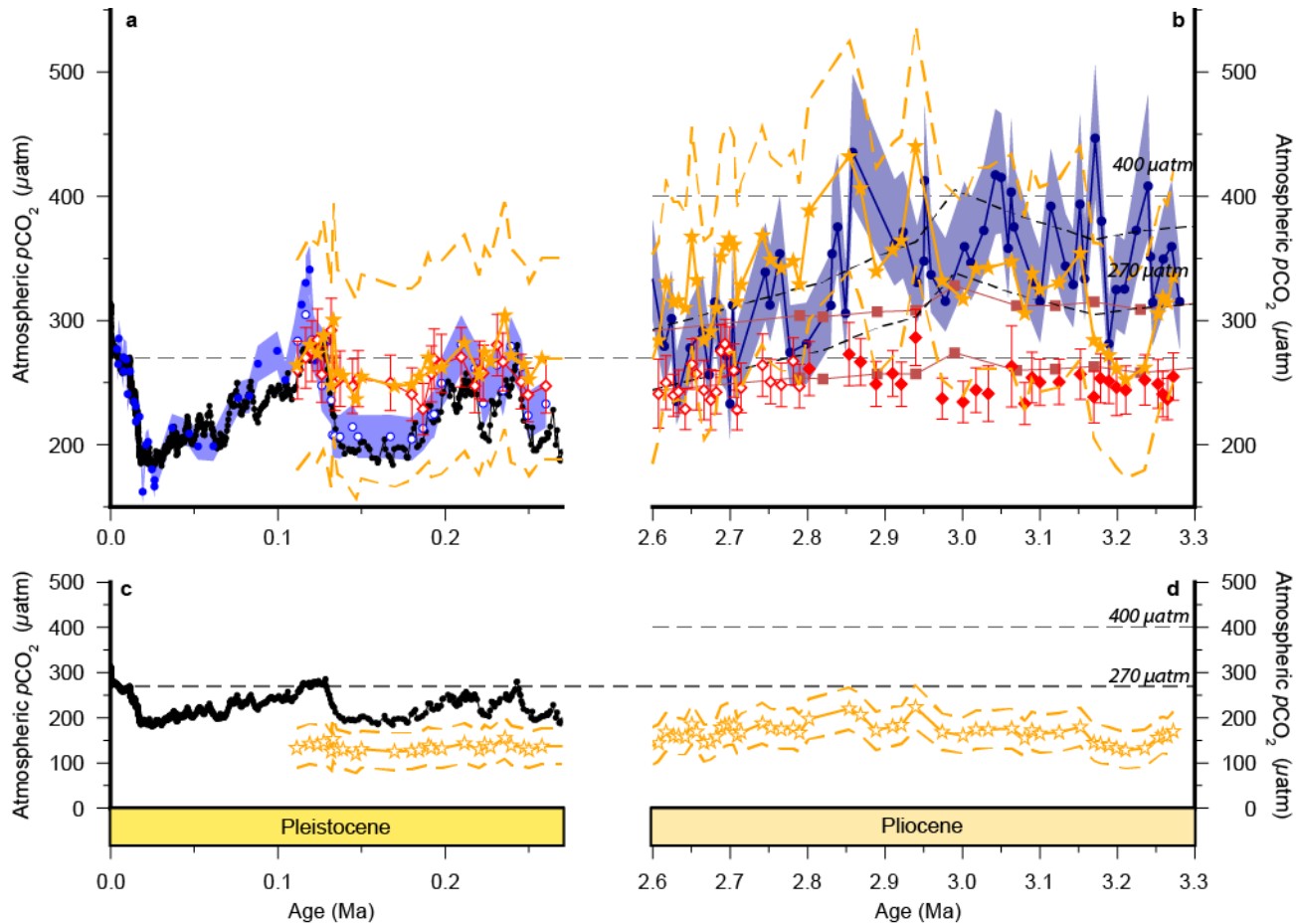

**Figure 6 Cell size corrections to CO2(εp-alk).** Ornamentation is the same as Figure 1, with the addition of cell size corrected CO2(εp-alk). The smaller liths than modern *E. huxleyii* across all of our records of our records mean that a direct application of the method of Henderiks and Pagani (2007) result in substantially lower CO2(εp-alk) throughout (orange open stars, panels c,d). As the main interest is in the effect of the Plio-pleistocene change in cell size we observe, we adjusted the b' term so that CO2(εp-alk) matched our uncorrected CO2(εp-alk) record during the last interglacial (orange filled stars, panals a,b).

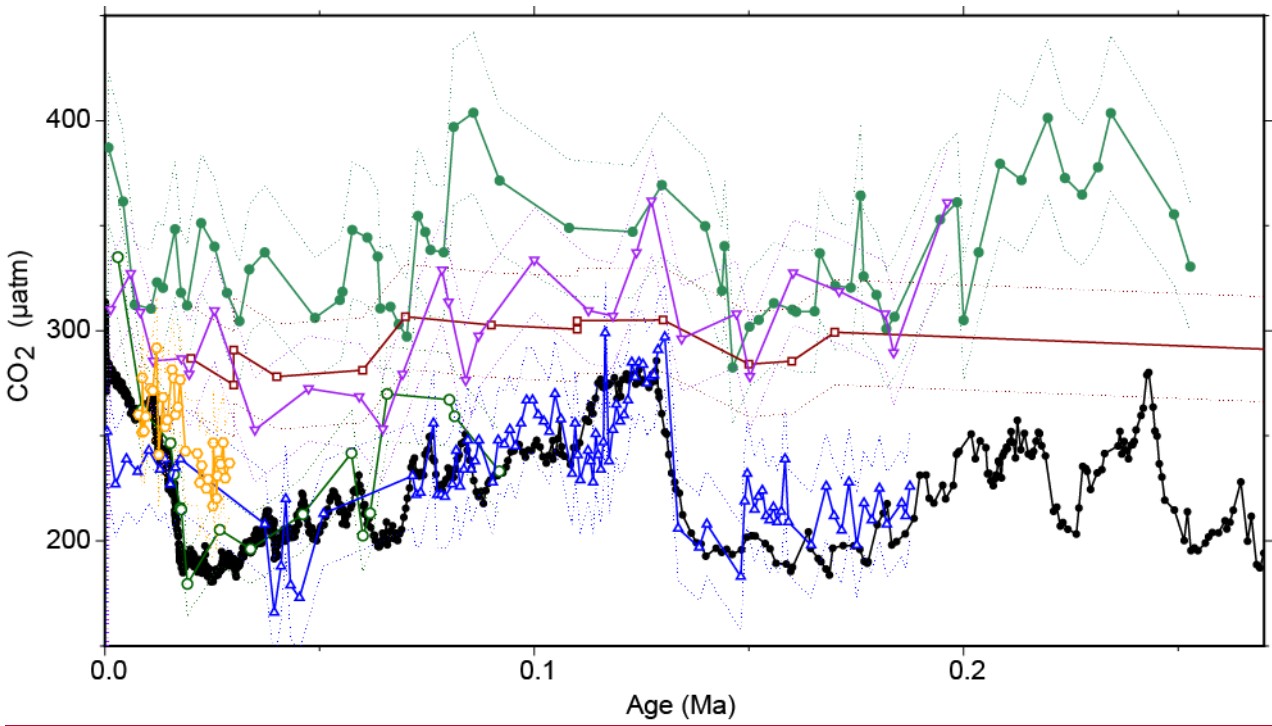

**Figure 7~~6~~** Recalculated CO$_{2(\epsilon p\text{-alk})}$. **Previous work** (Jasper et al., 1994; Jasper and Hayes, 1990) **calculated CO$_{2(\epsilon p\text{-alk})}$ using a different model; here we recalculate the earlier work using the modern methodology and Monte Carlo propagation applied to our other sites.** **All previous records have been recalculated using the same methodology as our new record, with some corrections and adjustments (for example for growth rate or lith size) removed to allow direct comparisons. Records are from MANOP Site C from the central equatorial Pacific (0° 57.2' N, 138° 57.3 W; green filled circles;** Jasper et al., 1994)**, DSDP Site 619 in the Pigmy Basin, northern Gulf of Mexico (27° 11.6'N, 91°24.5'W; green open circles** Jasper and Hayes, 1990)**, site 05PC-21 from the Japan Sea (38.40 °N, 131,55 °E; blue open triangles,** ~~(~~Bae et al., 2015)**, site NIOP464 in the Arabian Sea (22.15 °N, 63.35 °E; orange open hexagons** ~~(~~Palmer et al., 2010)**, site GeoB 1016-3 in the Angola Current (11.59 °S, 11.70 °E, purple inverted triangles** ~~(~~Andersen et al., 1999) **and ODP Site 925 (4° 12.25' N. 43°,29.33' W; dark red open squares,** (Zhang et al., 2013)**.~~MANOP Site C from the central equatorial Pacific (0° 57.2' N, 138° 57.3 W) is shown as green filled squares~~circles~~, DSDP Site 619 in the Pigmy Basin, northern Gulf of Mexico (27° 11.6'N, 91°24.5'W) is shown as open squares, and ice~~ Ice core data are shown as filled black circles and lines** (Bereiter et al., 2015 and Table 1)**. Dashed lines are 2σ uncertainties are from Monte Carlo error propagation as described elsewhere in the text. ~~Neither coccolith nor growth rate corrections were applied.~~**

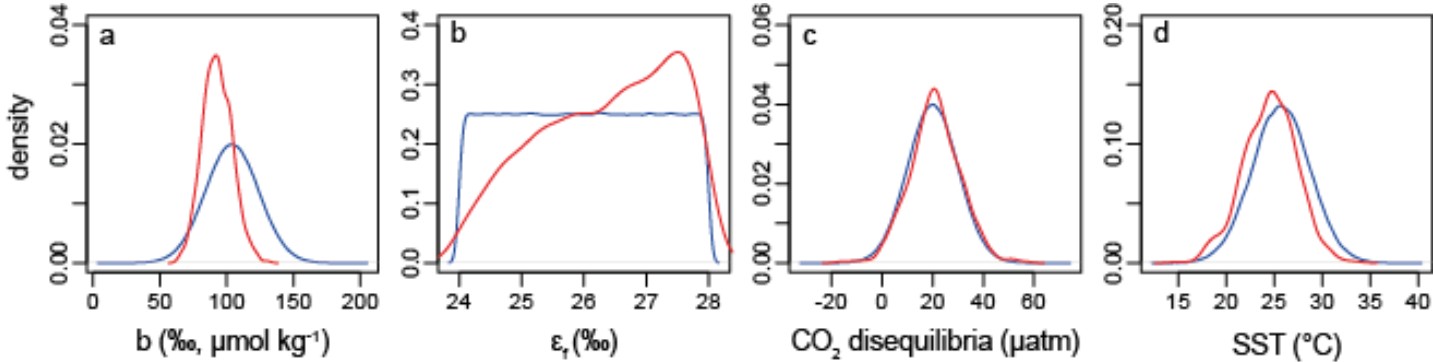

Figure 87. Example of the Bayesian treatment of the $CO_{2(\varepsilon p\text{-alk})}$ proxy, the sample shown is 0.15 Ma from ODP Site 999.  In all panels the prior is shown in blue and the posterior in red.  (a) the b-term, (b) $\varepsilon_f$, (c) the extent of $CO_2$ disequilibria, (d) sea surface temperature.

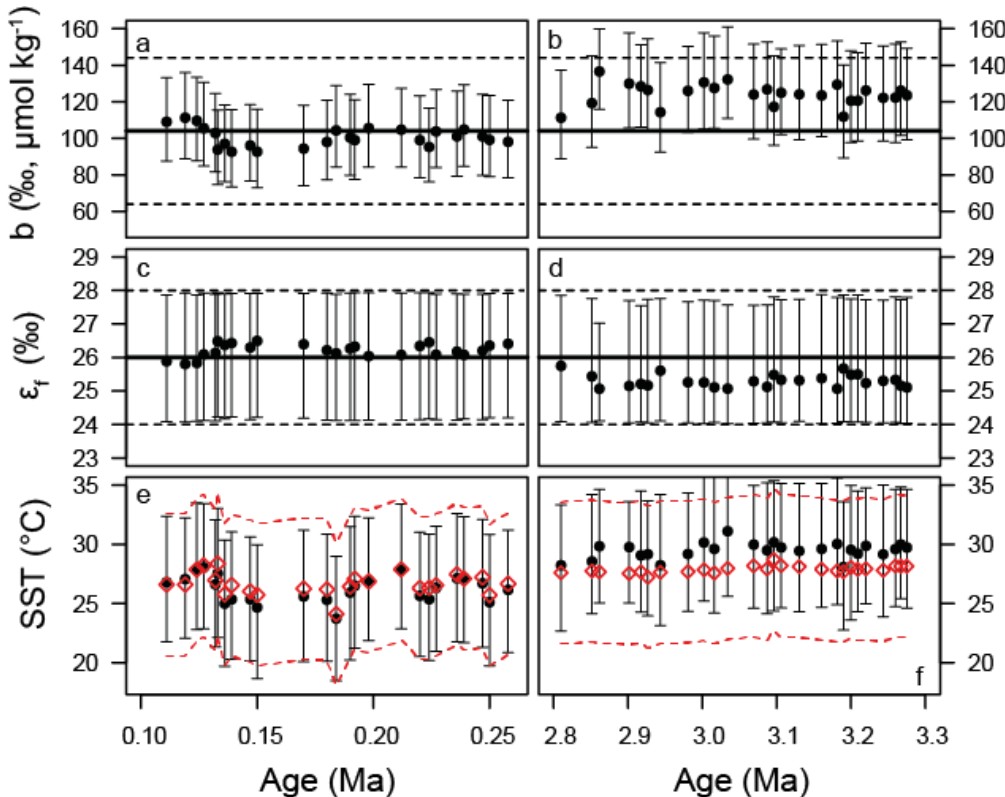

Figure 98. Timeseries of priors and posteriors for the b-term (a, b), $\varepsilon_f$ (c,d) and SST (e,f). The Pleistocene is shown on the panels on the left and the Pliocene on the right. In panels a-d the mean of the prior distribution is shown as a thick black line. For the b-term 95% of the input distribution is shown as a dotted line, for $\varepsilon_f$ the total range is shown. See Figure 7 for examples of these distributions as probability functions. For SST (e,f) the prior is shown as red diamonds with 95 % of the distribution shown as the dashed lines. In all panels the median of the posterior distributions are shown as circles with error bars encompassing 95 % of the range.

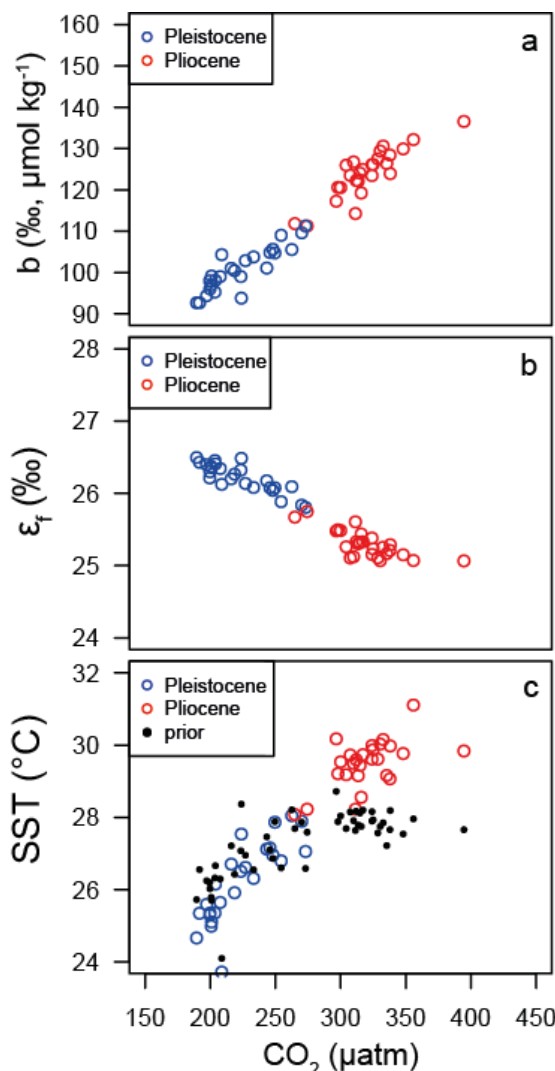

**Figure 109. Relationships between CO₂ and (a) b-term, (b) εf and (c) SST.  In each panel the median of each posterior distribution is shown in red for the Pliocene and blue for the Pleistocene.  Note that the CO₂ for each data point is either from the ice core or CO₂(δ11Bplank) for the Pleistocene and Pliocene, respectively. The linear patterns that emerge here essentially represent the relationships of the** Bidigare et al., (1997) **approach given our otherwise invariant εp.**

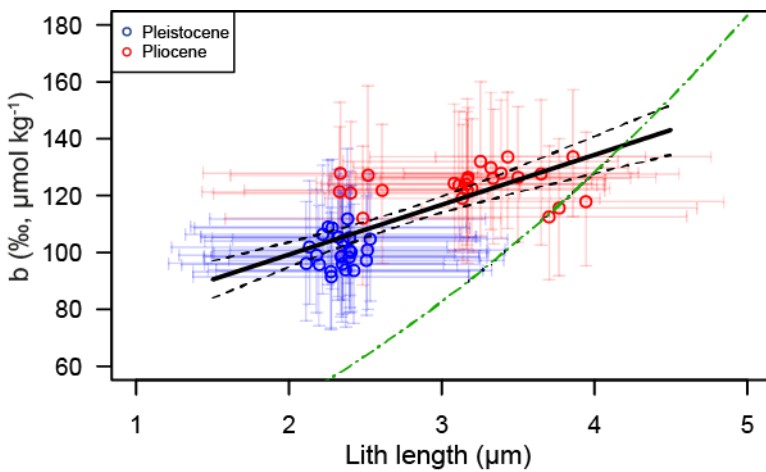

**Figure 11. Comparison between the posterior distribution for the Pliocene (red circles) and Pleistocene (blue circles) and the lith size correction of** (Henderiks and Pagani, (2007) **(green dot-dash line).**