# Peer review of "Insensitivity of alkenone carbon isotopes to atmospheric CO2 at"

_Climate of the Past, 2018_

## Referee Comment (RC1) · Anonymous Referee #1 · 5 Dec 2018

Badger et al. present an examination of the CO2 proxy derived from the stable carbon isotope composition of alkenones in marine sediments, assessing the potential factors driving apparent differences in the sensitivity of this proxy to CO2 under different climate conditions. The accurate and precise reconstruction of atmospheric CO2 is of critical importance for climate studies, because it allows us to consider in detail the overall radiative forcing of the atmosphere, the global scale response in terms of climate sensitivity, and in turn to identify regions or systems of sensitivity in terms of their response to CO2 and/or possible feedbacks driving CO2 variations. Previous investigations have proved valuable for providing information on global climate sensitivity to CO2 forcing beyond that which can be determined from the instrumental record or model experiments. It is therefore important that the highest quality CO2 data can be

generated, and this manuscript is a very valuable contribution.

As outlined clearly by Badger et al., there is a history of two different proxy systems being applied. One based on boron isotopes in foraminifer calcite, and the second utilising the known isotopic fractionation which occurs in alkenones during coccolithophore production, and which has been shown to be related (in part) to aqueous $CO_2$ concentrations. A long standing challenge for these two proxies has been the observed discrepancy in the absolute values of atmospheric $CO_2$ which are recorded, especially during the Pliocene and Pleistocene. As outlined here, the close alignment of recent boron-derived $CO_2$ measurements with ice core $CO_2$ data now prompts detailed examination of the alkenone proxy, to understand why this seems insensitive to the large fluctuations in late Pleistocene glacial-interglacial cycles. Badger et al. undertake this testing here, and also for the late Pliocene ($\sim$2.8 Ma). Their approach is well designed: they avoid issues around comparing samples from different locations/times by specifically applying both proxy approaches to the same sediment samples. They also push the data further than simply making descriptive, visual comparisons: by using the ice core $CO_2$ data as their target, they use Bayesian analysis to explore whether the parameters used in their calculations can account for some of the discrepancies. This is a really valuable part of the manuscript. The boron isotope data has already been published, and whilst some of the alkenone data has previously been published (>3.0 Ma), both a larger detailed data set and interesting statistical analyses are presented for the first time here to allow the authors to thoroughly tackle the stated problem.

The authors identify two main issues: (1) $CO_2$ reconstructions from alkenones record interglacial values but not glacial values for the late Pleistocene; (2) $CO_2$ reconstructions from alkenones in the late Pliocene show little difference to those of the late Pleistocene (despite boron estimates suggesting higher Pliocene $CO_2$). There is therefore some insensitivity of the alkenone $CO_2$ proxy to known variations in $CO_2$, which requires a re-examination of this proxy. Their Bayesian analysis highlights a potential influence from the SST calculations (for the Pliocene data) as well as potential flaws in

the way that physiological factors in the alkenone producers are accounted for. These are critical findings, and suggest that with our knowledge as it stands, alkenone stable carbon isotopes should not be applied as a proxy for CO2.

Overall the manuscript is generally well written (some minor typos), with high quality figures. It is clear that further and intensive work is required to address the concerns here, but the authors make some logical suggestions about areas to be targetted, drawing on a range of literature in support.

My main concerns rest with a need for some increased detail in the Introduction (to strengthen the need to tackle the question they pose) and Methods (at times it feels that the details are rather swiftly handled). I also have some questions around the discussion of differences in the signals recorded on glacial-interglacial timescales and longer term (Pleistocene-Pliocene). My suggestions are provided in more detail below, alongside some minor corrections (typos etc.):

(1) The Introduction (page 2) makes many statements about 'low' and 'high' CO2 worlds, but no numbers are given. It would be useful to give this context, considering that an expected audience might span Quaternary scientists (for whom an interglacial CO2 might be 'high') and those interested in Cenozoic climate evolution. Either state the known ranges where descriptive terms are used, or consider tabulating some of the studies you cite. Likewise, on page 2 lines 20-21 there is a note to different earth system sensitivity studies and their 'differences', but it isn't clear whether these are (in)significant, within error etc. Adding some of these details would really help the reader to get a quick sense of whether the problem posed here (different CO2 from different proxies) is something of major importance, or more nuanced and perhaps less critical.

(2) Methods detail. The methods section is well written, but on several occasions some details are missing which would help the reader to follow the flow and/or to understand the rationale for why certain approaches were taken. Specifically:

[Figure]

a. page 4 line 11: which alkenone-SST calibration was used? what was the value of modern salinity? The same as stated later for boron isotopes?

b. page 4 line 14-15: what were the instrument conditions for the d13C analysis? (GC method, IRMS conditions, reference gases or standards used for d13C ...). Or, less ideally, make reference to the Badger et al. (2013) study but only if the full instrument conditions are clearly stated there.

c. page 4 line 14-15: You state where the boron measurements were taken - for consistency can you also say where the d13C was measured?

d. page 5 line 5: this range and uncertainty for UK37 and calcite d13C are not explained. Given known non-linearity in the SST calibration at the upper end, and the notes of analytical variation in generating SSTs in this study (line 7 on this page), can the authors state whether this range of uncertainty is more than required, or is it instead a realistic estimate when different calibrations and replicates are considered? Later in the manuscript there is a suggestion that 'realistic' values are the focus here, but that is not obvious from this paragraph.

e. page 5 line 10: state how was the disequilibrium was accounted for, even if it is just a simple step.

f. page 6 line 1: confirm whether these are the same salinity values used for the alkenones (see comment (a) above).

g. Page 6 line 5: clarify whether these 'Values from Foster' are the reconstructed CO2, or something else (the inorganic chemical constants from the previous sentence?).

(3) Discussion of different timescales. On page 7 (line 17) the authors state that the pCO2 record is "largely stable and invariant ... through both the Pliocene and PLeistocene...". But in Figures 3 and 4 I would argue that there is still variability on orbital (glacial-interglacial timescales) which can be identified in the original alkenone d13C data as well as in the reconstructed pCO2. The apparent stability and lack of variance

is instead reflected in the comparison of the longer term data sets i.e. between the Pliocene and Pleistocene, but only for pCO2 (all other measurements do seem to show an offset). This statement on page 7 (line 17) requires some expansion to account for the differences in temporal response i.e. whether there is "a lack of variability". What now becomes intriguing is that not only does pCO2 fail to record Pleistocene glacials, but apparently also Pliocene interglacials, despite offsets being determined in the original alkenone d13C and epsilon-p values.

a. page 10 line 24: notes to the possible influence of an incorrect SST calculation on the CO2 calculation. Here, it would be useful to reflect on the uncertainty range used in the original calculations. Would using a different SST calibration yield a better result, especially given the known non-linearities in UK37-SST calibrations at high SSTs?

b. page 11 paragraph 2: parameterisation of physiological factors. The authors note that the dominant alkenone-producing species today is Emiliania huxleyi. But, given the importance of physiological factors suggested here, is it known that E hux has always been present/dominant at this site through the Pleistocene glacial-interglacial cycles? (the cited paper by Winter et al., 2002 is for modern seawater) Is it known which other species might contribute, and if they are closely related? For the Pliocene, this becomes perhaps more crucial: the manuscript does not highlight that in the absence of E hux, there must be a different set of producers in the Pliocene (which could perhaps account for a different relationship between epsilon-p and CO2?). Is there any information from this site about which coccolithophore species are present in the Pliocene and through the Pleistocene? If there isn't, then it would still be useful to state this as an uncertainty. Have the authors looked at alkenone accumulation rates as a possible indicator of export flux (and potentially productivity) to see if there are any glacial-interglacial or Pliocene-Pleistocene differences?

Minor corrections

- Format UK37' with the correct sub/superscript throughout

[Figure]

- page 7 line 16 – comment about glacial temperatures: no comment made about the unusually cold SST which isn't in the glacial

- page 6 line 11 – explain how "using alkenones limits the variation of cell geometry"... aren't alkenones synthesised by multiple species of haptophyte, which could have different cell geometries? Any citation for support?

- page 7 lines 18-20 – this information about estimating 'b' seems out of place here, and interrupts the flow. Better in the methods section, or in the paragraph which follows where the differences to Seki can be outlined?

- page 10 line 15 – "and test which variables maybe responsible" ?

- page 10 line 16 – "largely invariant for the Pleistocene ..."?

- add analytical uncertainty bars to Figure Panels 3a and 3e (or state if these are too small to be seen)

- why are figures 3 and 4 showing different width of time for pliocene? Please can figure 3 show the pliocene data as well (which is shown in figures 2 and 4?)

- caption figure 4 – isnt lith size panels a and d? Clarify that the drop in lith size (page 9 lines 12-15) is reflected in mean/median (figure 4?), since there doesn't seem to be any shift in the ranges between the two time intervals.

---

## Referee Comment (RC2) · Anonymous Referee #2 · 17 Jan 2019

This manuscript presents new alkenone-based pCO2 data along with previously published 11B-based pCO2 reconstructions from the same late-Pleistocene and Pliocene samples. This is a timely comparison and tackles the question of how well the alkenone CO2 proxy as currently applied can reconstruct changes in CO2 near modern pCO2 levels. In an interesting exploration of the alkenone pCO2 data, the authors present novel estimates for the input parameters that would allow the proxy to reconstruct known pCO2 values. This exploration highlights some of the fundamental problems facing the alkenone pCO2 proxy in its current form and will hopefully spark new work to better understand carbon isotopic fractionation by coccolithophore algae and its relationship to CO2 concentrations. The manuscript is well written, novel, and will be of interest to a wide variety of scientists. The figures are generally easily interpreted and

clear. There are not any major issues with the manuscript and for this reason I suggest to publish with minor revisions. Comments and suggestions for these revisions are described below.

My only substantial comment is that I think the authors should calculate the Pliocene pCO2 values from the alkenone proxy with the cell size corrections added (and do the same for the Pleistocene samples for comparison). I realize this doesn't change the orbital-scale insensitivity in the Pleistocene. But, it does allow comparison of the absolute value and magnitude of change between the Pleistocene and Pliocene windows in both proxies. It will also change the posterior distributions of the input variables for the alkenone pCO2 reconstructions. My sense is that it will bring the b-parameter, ef, and SST posteriors more in line with the priors. The lith size changes are on the order of 150 to 200% higher in the Pliocene with respect to the Pleistocene (it appears from the figure). That is substantial and would increase the estimated Pliocene pCO2 values into the high 300 to low 400 ppm range – very similar to the 11B pCO2 estimates. This may indicate that the alkenone pCO2 proxy agrees in magnitude with Plio-Pleistocene pCO2 changes and thus may be sensitive at higher CO2 levels but not at the very low Pleistocene glacial levels. The authors suggest this might be the case in the conclusions. If they show it is the case with their Pliocene reconstructions it would provide some nice empirical support (and they should mention this in the abstract).

Page 4 line 15 – how were alkenone 13C isotope measurements calibrated and what was the replicate precision and the accuracy (i.e. uncertainty from analysis plus uncertainty in realizing the VPDB scale). (Same comment for p5 line 6).

Page 9 line 20 – The previous paragraph stated that there is some evidence for a reduction in productivity during glacials and if that translates to cell-specific growth rates then it could explain some of the lack of signal in the alkenone pCO2 reconstructions. In light of that observation, the following statement is confusing to me: "This suggests that either our understanding of growth rate effects on CO2($\varepsilon$p-alk) is incorrect, or the estimation of cell size using preserved liths does not capture original cell size

variations. . ..." Doesn't the prior statement suggest that our understanding of growth rate effects may actually be correct?

Page 10 line 25 – This paragraph is quite instructive, nice! One question is how the SST posterior is calculated? For the Pliocene, the SST would also affect the pCO2 estimated by the 11B method. Thus, if one assumes a pCO2 from the 11B and then gets a posterior SST from the alkenone proxy, this different SST would change the 11B pCO2 estimate and thus the alkenone SST posterior based upon the earlier 11B CO2 value is no longer correct.

Page 11 line 5 – ". . .the current understanding of the CO2(ep-alk) proxy is wanting." Yes, the b term may not in fact capture the scaling of physiological parameters or the truly important parameters.

Page 11 line 19 – The sentence starting, "Additionally, CO2 optima. . ." is a bit unclear. I think the authors are saying that different species have different CO2 optima so that CCM effects may vary between regions where different species dominate? But maybe not that. Please rewrite and clarify.

Very nice, concise explication of the issues and a nice manuscript!

———————————————

---

## Author Comment (AC1) · 21 Jan 2019

We are grateful to the referee for their comments and for their kind words on the manuscript. We are happy to implement all the suggestions made by the referee in a revised manuscript, including increasing the level of detail we provide in the methods, and clarifying our language surrounding 'low' and 'high' levels of CO$_2$. In this context we use 'low' to mean the sub-300 ppm value of atmospheric CO$_2$ measured for the late Pleistocene and Holocene, and use 'high' to mean CO$_2$ values above 300 ppm, which most proxy records suggest were present for much of the pre-Pleistocene Cenozoic.

---

## Author Comment (AC2) · 21 Jan 2019

We are grateful to the referee for their comments and for their kind words on the manuscript. We are happy to make the minor revisions the referee suggests in a revised manuscript. Specifically we are happy to include a further figure at the appropriate point in the discussion which shows $pCO_2$ calculated with a lith-size correction included, and to include additional posterior distributions which take account of this correction.

To briefly answer the comment about growth rates (page 9 line 20) our point in the previous paragraph is that growth rates could be part of the story (if the evidence of low productivity is correct) but that the scale of change required the move the reconstructed $CO_{2(\epsilon p-alk)}$ in line with the $CO_{2(\delta 11Bplank)}$ and ice core $CO_2$ for the glacial is very substantial (see Figure 1), and greater than we think likely based on our current understanding of how growth rates would effect $\epsilon_p$. This suggests to us that if growth rate is the main cause for the discrepancy our understanding of growth rate effects on $\epsilon_p$ is incorrect. We are happy to clarify this in a revised manuscript.

[Figure]

[Figure]

**Fig. 1.** Only if growth rate (a) is modelled to be an order of magnitude lower in the glacial can alkenone CO2 (b; orange diamonds) match ice core (b; black circles) and boron CO2 (b; blue, Chalk et al 2017)

---

## Author Response (AR1)

Reviewer 1

> *"My main concerns rest with a need for some increased detail in the Introduction (to strengthen the need to tackle the question they pose) and Methods (at times it feels that the details are rather swiftly handled). I also have some questions around the discussion of differences in the signals recorded on glacial-interglacial timescales and longer term (Pleistocene-Pliocene). My suggestions are provided in more detail below, alongside some minor corrections (typos etc.)"*

We have increased the detail in the introduction (Page 2 lines 12-14) and methods (page 4 lines 18-4, page 5 lines 1-8, page 6 lines 16-17, lines 19-21, line 23, page 7 lines 10-17) and tightened up our use of language surrounding timescales throughout the manuscript.

> *"(1) The Introduction (page 2) makes many statements about 'low' and 'high' CO2 worlds, but no numbers are given. It would be useful to give this context, considering that an expected audience might span Quaternary scientists (for whom an interglacial CO2 might be 'high') and those interested in Cenozoic climate evolution. Either state the known ranges where descriptive terms are used, or consider tabulating some of the studies you cite."*

We now specific enumerate $CO_2$ levels in the introduction (Page 2 lines 12-14) and have tightened up our language on $CO_2$ levels throughout the manuscript.

> *"Likewise, on page 2 lines 20-21 there is a note to different earth system sensitivity studies and their 'differences', but it isn't clear whether these are (in)significant, within error etc. Adding some of these details would really help the reader to get a quick sense of whether the problem posed here (different CO2 from different proxies) is something of major importance, or more nuanced and perhaps less critical."*

We've added a further sentence here which now explains the discrepancy (Page 2 lines 21-24)

> *(2) Methods detail. The methods section is well written, but on several occasions some details are missing which would help the reader to follow the flow and/or to understand the rationale for why certain approaches were taken. Specifically:*
>
> *a. page 4 line 11: which alkenone-SST calibration was used? what was the value of modern salinity? The same as stated later for boron isotopes?*
>
> *b. page 4 line 14-15: what were the instrument conditions for the d13C analysis? (GC method, IRMS conditions, reference gases or standards used for d13C ...). Or, less ideally, make reference to the Badger et al. (2013) study but only if the full instrument conditions are clearly stated there.*
>
> *c. page 4 line 14-15: You state where the boron measurements were taken – for consistency can you also say where the d13C was measured?*

We have greatly expanded the Methods section and now include all of these details (Page 4 lines 18 – 24 and 5 lines 1-8).

> *d. page 5 line 5: this range and uncertainty for UK37 and calcite d13C are not explained. Given known non-linearity in the SST calibration at the upper end, and*

*the notes of analytical variation in generating SSTs in this study (line 7 on this page), can the authors state whether this range of uncertainty is more than required, or is it instead a realistic estimate when different calibrations and replicates are considered? Later in the manuscript there is a suggestion that 'realistic' values are the focus here, but thatis not obvious from this paragraph.*

We have added in an explanatory note about the alkenones earlier in the methods (Page 5 lines 5-7) the calcite d13C uncertainty is a generous estimate based on analytical uncertainty of the measurements.

> *e. page 5 line 10: state how was the disequilibrium was accounted for, even if it is just a simple step.*

This is just a simple subtraction, which we now include in the methods (Page 6 lines 1-2)

> *f. page 6 line 1: confirm whether these are the same salinity values used for the alkenones (see comment (a) above).*

It is, and this is now included (Page 6 line 17, Page 4 line 14)

> *g. Page 6 line 5: clarify whether these 'Values from Foster' are the reconstructed CO2, or something else (the inorganic chemical constants from the previous sentence?).*

These are the reconstructed $CO_2$ values, which we now state (Page 6 line 23)

> *(3) Discussion of different timescales. On page 7 (line 17) the authors state that the pCO2 record is "largely stable and invariant ... through both the Pliocene and PLeistocene...". But in Figures 3 and 4 I would argue that there is still variability on orbital (glacial-interglacial timescales) which can be identified in the original alkenone d13C data as well as in the reconstructed pCO2. The apparent stability and lack of variance is instead reflected in the comparison of the longer term data sets i.e. between the Pliocene and Pleistocene, but only for pCO2 (all other measurements do seem to show an offset). This statement on page 7 (line 17) requires some expansion to account for the differences in temporal response i.e. whether there is "a lack of variability". What now becomes intriguing is that not only does pCO2 fail to record Pleistocene glacials, but apparently also Pliocene interglacials, despite offsets being determined in the original alkenone d13C and epsilon-p values.*

Although there is variability in the alkenone $\delta^{13}C$ in the Pliocene and the Pleistocene it is not on orbital timescales, and as can be seen in Figures 3 and 4 much of it is within error (especially once it is converted to $CO_2$). While there are wiggles in the data, these do not match known glacial-interglacial cycles in the Pliocene, and whilst there is a positive slope in Figure 4b, it is not statistically different from zero, as our analysis shows.

> a. *page 10 line 24: notes to the possible influence of an incorrect SST calculation on the CO2 calculation. Here, it would be useful to reflect on the uncertainty range used in the original calculations. Would using a different SST calibration yield a better result, especially given the known non-linearities in UK37-SST calibrations at high SSTs?*

We have added a sentence here (page 12 line 14-16) noting that if there is a SST error it would need to be substantial to go beyond what we include in the uncertainty propagation. The Mg/Ca record from Chalk et al., 2017 does not suggest that our alkenone SST is significantly muted (although likely recording a slightly different part of the water column).

> *b. page 11 paragraph 2: parameterisation of physiological factors. The authors note that the dominant alkenone-producing species today is Emiliania huxleyi. But, given the importance of physiological factors suggested here, is it known that E hux has always been present/dominant at this site through the Pleistocene glacial-interglacial cycles? (the cited paper by Winter et al., 2002 is for modern seawater) Is it known which other species might contribute, and if they are closely related? For the Pliocene, this becomes perhaps more crucial: the manuscript does not highlight that in the absence of E hux, there must be a different set of producers in the Pliocene (which could perhaps account for a different relationship between epsilon-p and CO2?). Is there any information from this site about which coccolithophore species are present in the Pliocene and through the Pleistocene? If there isn't, then it would still be useful to state this as an uncertainty. Have the authors looked at alkenone accumulation rates as a possible indicator of export flux (and potentially productivity) to see if there are any glacial-interglacial or Pliocene-Pleistocene differences?*

*Emiliania huxleyi* coccoliths are first recognised in the fossil record at around 290 Ka, and take over as the dominant species in the ocean at around 82 Ka (Raffi et al., 2006) we now include this reference which was missed out), prior to this the closely related *Gephyrocapsa oceanica* (or related species) dominated, and was likely the major alkenone producer (Raffi et al., 2006). Most of our Pleistocene record therefore probably represents alkenones produced by *Gephyrocapsa* spp., whilst the Pliocene record is likely another closely related noelaerhabdaceae such as *Reticulofenestra*. We cannot therefore rule out these different species having different species having different physiological responses (although note that they are very closely related species of the same family and likely occupied a similar niche) and now include a sentence to that effect (page 13 lines 10-17). We do however note that the nearby Site 925 record is similarly invariant through the last glacial-interglacial cycle, and would have been produced entirely by *E. huxleyi*, and so this change in species cannot be the complete explanation.

> *Minor corrections*
>
> *- Format UK37' with the correct sub/superscript throughout*

Done.

> *- page 7 line 16 – comment about glacial temperatures: no comment made about the, unusually cold SST which isn't in the glacial*

This is likely just an outlier and so we have chosen not to discuss it in detail.

> *- page 6 line 11 – explain how "using alkenones limits the variation of cell geometry"... aren't alkenones synthesised by multiple species of haptophyte, which could have different cell geometries? Any citation for support?*

Using alkenones rather than other biomarkers or bulk organic matter to calculate $\varepsilon_p$ restricts source organisms to the very limited group of haptophytes that produce alkenones, this excludes organic

matter produced by organisms such as diatoms, radiolarians and dinoflagelates (for example) which have a much greater range of cell geometries. This is covered in Popp et al., (1998) and Laws et al., (1997) and we have now moved that reference from earlier in the sentence to make that clear (page 7 line 4)

> *- page 7 lines 18-20 – this information about estimating 'b' seems out of place here, and interrupts the flow. Better in the methods section, or in the paragraph which follows where the differences to Seki can be outlined?*

This is already included in the methods section (page 5 line 15) so we have removed the sentence here to improve the flow.

> *- page 10 line 15 – "and test which variables maybe responsible" ?*

Done (page 12 line 4)

> *- page 10 line 16 – "largely invariant for the Pleistocene …"?*

Done (page 12 line 5)

> *- add analytical uncertainty bars to Figure Panels 3a and 3e (or state if these are too small to be seen)*

Done.

> *- why are figures 3 and 4 showing different width of time for pliocene? Please can figure 3 show the pliocene data as well (which is shown in figures 2 and 4?)*

These figures show the new data, the remainder of the Pliocene data was published in Badger et al., (2013).

> *- caption figure 4 – isnt lith size panels a and d? Clarify that the drop in lith size (page 9 lines 12-15) is reflected in mean/median (figure 4?), since there doesn't seem to be any shift in the ranges between the two time intervals.*

The caption and sentence (page 10 line 19) has been changed.

Reviewer 2

*My only substantial comment is that I think the authors should calculate the Pliocene pCO2 values from the alkenone proxy with the cell size corrections added (and do the same for the Pleistocene samples for comparison). I realize this doesn't change the orbital-scale insensitivity in the Pleistocene. But, it does allow comparison of the absolute value and magnitude of change between the Pleistocene and Pliocene windows in both proxies. It will also change the posterior distributions of the input variables for the alkenone pCO2 reconstructions. My sense is that it will bring the b-parameter, ef, and SST posteriors more in line with the priors. The lith size changes are on the order of 150 to 200% higher in the Pliocene with respect to the Pleistocene (it appears from the figure). That is substantial and would increase the estimated Pliocene pCO2 values into the high 300 to low 400 ppm range – very similar to the 11B pCO2 estimates. This may indicate that the alkenone pCO2 proxy agrees in magnitude with Plio-Pleistocene pCO2 changes and thus may be sensitive at higher CO2 levels but not at the very low Pleistocene glacial levels. The authors suggest this might be the case in the conclusions. If they show it is the case with their Pliocene reconstructions it would provide some nice empirical support (and they should mention this in the abstract).*

We chose not to build the lith size correction into our Bayesian approach because the correction, following Henderiks and Pagani (2007) and Seki et al. (2010) involves some rather arbitrary scaling. This is because the correction results in very low $CO_2$ during the Pleistocene compared to the ice core (Figure 6 panel c). If we follow Seki et al. (2010) and scale b' so that the ice core interval is correct and then apply this same scaling of b' to the Pliocene interval then, as the reviewer predicted, we do see a good overlap between the boron and the alkenone (new Figure 6). Although it should be noted that this correction does not induce glacial-interglacial cyclicity in the Pleistocene alkenone data. To more empirically examine if cell-size played a role in the differences as the reviewer suggests, but to also avoid the arbitrary nature of the cell-size correction which is difficult to enact in a Bayesian sense, we now include a figure showing lith size vs. posterior b (Figure 11). This shows that there is a good correlation between b and lith size ($R^2$ = 0.52, p<<0.01) but it is quite different to the relationship predicted by Hendriks and Pagani (2007) shown by the green dot-dash line in the Figure. This suggests that if cell-size is important, its influence on b is not as we currently understand it. We now add these two figures figure and text to this effect in the revised manuscript (Figure 6, Figure 11, page 7 lines 10-17, page 13, lines 1-6).

*Page 4 line 15 – how were alkenone 13C isotope measurements calibrated and what was the replicate precision and the accuracy (i.e. uncertainty from analysis plus uncertainty in realizing the VPDB scale). (Same comment for p5 line 6).*

We've now substantially expanded the methods section (page 4 lines 20-24, page 5 lines 1-8) and include this information there.

*Page 9 line 20 – The previous paragraph stated that there is some evidence for a reduction in productivity during glacials and if that translates to cell-specific growth rates then it could explain some of the lack of signal in the alkenone pCO2 reconstructions. In light of that observation, the following statement is confusing to me: "This suggests that either our understanding of growth rate effects on CO2("p-alk) is incorrect, or the estimation of cell size using preserved liths does ot*

*capture original cell size variations: : :.”* Doesn't the prior statement suggest that
*our understanding of growth rate effects may actually be correct?*

Our point in the previous paragraph is that growth rates could be part of the story (if the evidence of low productivity is correct) but that the scale of change required the move the reconstructed $CO_{2(ep\text{-}alk)}$ in line with the $CO_{2(\delta11Bplank)}$ and ice core $CO_2$ for the glacial is very substantial (see Figure 1), and greater than we think likely based on our current understanding of how growth rates would effect $\varepsilon_p$, This suggests to us that if growth rate is the main cause for the discrepancy our understanding of growth rate effects on $\varepsilon_p$ is incorrect. We have added text to explain this (page 11 line 5-6).

> *Page 10 line 25 – This paragraph is quite instructive, nice! One question is how*
> *the SST posterior is calculated? For the Pliocene, the SST would also affect the*
> *pCO2 estimated by the 11B method. Thus, if one assumes a pCO2 from the 11B*
> *and then gets a posterior SST from the alkenone proxy, this different SST would*
> *change the 11B pCO2 estimate and thus the alkenone SST posterior based upon*
> *the earlier 11B CO2 value is no longer correct.*

This is potentially correct, but we chose not to do this for two reasons, 1) the temperature effect on $CO_{2(\delta11Bplank)}$ is actually fairly minor (less so than $CO_{2(ep\text{-}alk)}$) and 2) as we use different SST records for $CO_{2(ep\text{-}alk)}$ and $CO_{2(\delta11Bplank)}$ ($U_{37}^{K'}$ for the former and Mg/Ca in planktic foraminifera for the latter) to keep the carrier organisms the same, producing a sutiable SST for the $CO_{2(\delta11Bplank)}$ would be non-trivial.

> *Page 11 line 5 – “: : :the current understanding of the CO2(ep-alk) proxy is*
> *wanting.” Yes, the b term may not in fact capture the scaling of physiological*
> *parameters or the truly important parameters.*

Although this may not have been the reviewers intention, we have lifted this nice phrasing almost in it's entirety and added it at page 12 line 21-2, we hope they don't mind!

> *Page 11 line 19 – The sentence starting, “Additionally, CO2 optima: : :” is a bit*
> *unclear. I think the authors are saying that different species have different CO2*
> *optima so that CCM effects may vary between regions where different species*
> *dominate? But maybe not that. Please rewrite and clarify.*

We've restructured this sentence (page 14 lines 3-5) which is now hopefully clearer.

Additionally, following correspondence prompted by the publication of the discussion paper, we now improve the clarity and robustness of our citation of the ice core data with the addition of Table 1, and changed the discussion of other $CO_{2(ep\text{-}alk)}$ records from the Pleistocene to better represent the intent of some of those works (page 11 lines 11-20).

[revised manuscript text omitted]